# Altered NMDAR signaling underlies autistic-like features in mouse models of CDKL5 deficiency disorder

Sheng Tang[1,2,3,4], Barbara Terzic [1,4], I-Ting Judy Wang[1], Nicolas Sarmiento[1], Katherine Sizov[1], Yue Cui[1], Hajime Takano[2,3], Eric D. Marsh[2,3], Zhaolan Zhou[1] & Douglas A. Coulter[2,3]

CDKL5 deficiency disorder (CDD) is characterized by epilepsy, intellectual disability, and autistic features, and CDKL5-deficient mice exhibit a constellation of behavioral phenotypes reminiscent of the human disorder. We previously found that CDKL5 dysfunction in forebrain glutamatergic neurons results in deficits in learning and memory. However, the pathogenic origin of the autistic features of CDD remains unknown. Here, we find that selective loss of CDKL5 in GABAergic neurons leads to autistic-like phenotypes in mice accompanied by excessive glutamatergic transmission, hyperexcitability, and increased levels of postsynaptic NMDA receptors. Acute, low-dose inhibition of NMDAR signaling ameliorates autistic-like behaviors in GABAergic knockout mice, as well as a novel mouse model bearing a CDD-associated nonsense mutation, CDKL5 R59X, implicating the translational potential of this mechanism. Together, our findings suggest that enhanced NMDAR signaling and circuit hyperexcitability underlie autistic-like features in mouse models of CDD and provide a new therapeutic avenue to treat CDD-related symptoms.

[1] Department of Genetics, University of Pennsylvania Perelman School of Medicine, Philadelphia, PA 19104, USA. [2] Department of Neuroscience, Neurology, and Pediatrics, University of Pennsylvania Perelman School of Medicine, Philadelphia, PA 19104, USA. [3] The Research Institute of the Children's Hospital of Philadelphia, Philadelphia, PA 19104, USA. [4] These two authors contributed equally: Sheng Tang, Barbara Terzic. Correspondence and requests for materials should be addressed to Z.Z. (email: zhaolan@pennmedicine.upenn.edu) or to D.A.C. (email: coulterd@email.chop.edu)

Mutations in cyclin-dependent kinase-like 5 (*CDKL5*) cause a severe neurodevelopmental disorder characterized by early-onset seizures, intellectual disability, and autistic features[1–3]. Mouse models of CDD have recapitulated numerous aspects of the human disorder and, despite a subtle overall degree of impairment, they show behavioral deficits such as impaired learning, reduced sociability, motor dysfunction, and altered anxiety-related behaviors[4–7]. The extent to which these behavioral deficits arise from common or distinct mechanisms, however, remains unclear.

CDKL5 protein expression is enriched in the forebrain, primarily in glutamatergic and GABAergic neurons[4,8–11]. In glutamatergic neurons, CDKL5 has been found at the postsynaptic density, where it interacts with PSD-95 and NGL-1[12,13]. In addition, loss of CDKL5 both in vitro and in vivo is associated with changes in dendritic morphology, spine density, excitatory synaptic transmission, and synaptic plasticity[13–16]. Mice lacking CDKL5 selectively in forebrain glutamatergic neurons (Nex-cKO) show impaired learning and memory[14], reminiscent of the intellectual disability found in CDD. Interestingly, Nex-cKO mice did not show other behavioral deficits found in *Cdkl5* constitutive knockout mice, such as alterations in sociability, stereotypic behavior, locomotion, motor coordination, and anxiety-related behavior[14]. These findings imply that distinct, cell type-specific etiologies underlie CDD-related behavioral phenotypes in mice.

Mice lacking CDKL5 show numerous functional changes at the synaptic and circuit levels. A recent study found that *Cdkl5* constitutive knockout mice demonstrated increased NMDA-dependent synaptic transmission and enhanced long-term potentiation at hippocampal synapses[16]. In contrast, long-term potentiation is decreased in the somatosensory cortex of *Cdkl5* knockout mice[15]. Furthermore, selectively ablating CDKL5 from glutamatergic neurons leads to increased glutamatergic and GABAergic synaptic transmission, disrupted microcircuit dynamics, and learning and memory impairment[14]. While some of these differences are potentially attributable to different genetic backgrounds of mouse models of CDD, the differences between *Cdkl5* constitutive knockout mice and Nex-cKO mice suggest the existence of additional, non-glutamatergic mechanisms that may mediate CDD-related behavioral deficits. Notably, the function of CDKL5 in forebrain GABAergic neurons, where CDKL5 is also highly expressed, has yet to be elucidated[8].

Here, we selectively ablate CDKL5 expression in forebrain GABAergic neurons (Dlx-cKO). We found that these mice exhibit an autistic-like phenotype, but, in contrast to Nex-cKO mice, show preserved learning and memory[14]. In addition, Dlx-cKO mice show an enhancement of excitatory synaptic transmission and circuit-level hyperexcitability, coupled with elevated levels of NMDA receptors. Reducing NMDAR activity using an uncompetitive antagonist, memantine, significantly mitigated the behavioral deficits found in Dlx-cKO mice. To examine the translational potential of these findings, we generated a novel CDD model bearing a patient mutation, CDKL5 R59X, and found that these mice, similarly to Dlx-cKO mice, show an elevation of NMDA receptors. Importantly, acute, low-dose NMDAR blockade selectively ameliorates autistic-like features in this CDD model. Taken together, our findings support a novel mechanism by which CDKL5 loss in GABAergic neurons leads to excessive NMDAR signaling and contribute to the etiology of autistic-like behaviors in mouse models of CDD.

## Results

### CDKL5 GABAergic deletion results in autistic-like features.
Our previous findings showed that CDD-related learning and memory impairment has origins in forebrain glutamatergic

neurons in mice[14]. Given that CDKL5 is also highly expressed in forebrain GABAergic neurons, we generated conditional knockout mice selectively lacking CDKL5 in this cell population (Dlx-cKO) using the Dlx-5/6 Cre driver[17] (Supplemental Fig. 1A, B). Dlx-cKO mice showed normal growth and body weight through adulthood and no obvious physical abnormalities (Supplemental Fig. 2A). We then performed a battery of behavioral assays, similar to those in previous studies of *Cdkl5* constitutive KO[4] and Nex-cKO mice[14]. Compared to WT controls, Dlx-cKO showed no changes in locomotor activity, anxiety-related behaviors, and motor coordination (Supplemental Fig. 2B–D).

In contrast, Dlx-cKO mice demonstrated significantly reduced social interaction on the three-chamber social approach assay, showing diminished preference for investigating a social stimulus as compared to an object (Fig. 1a, b). When allowed the opportunity for direct interaction with a novel stimulus mouse, Dlx-cKO mice also spent significantly less time initiating contact in comparison to wild-type controls (Fig. 1c). To rule out the involvement of an olfactory deficit underlying reduced social preference, we conducted the olfactory habituation–dishabituation test. Dlx-cKO mice showed an intact ability to discriminate between different scents but spent reduced time sniffing a social scent, a feature consistent with a reduced interest in social stimuli (Fig. 1d).

In addition to social deficits, another defining feature of human autism spectrum disorders (ASDs) is the presence of repetitive or stereotypic behaviors. We next assessed repetitive behaviors in Dlx-cKO mice. In a home-cage like environment, Dlx-cKO mice showed significantly increased time engaging in stereotypic behaviors such as grooming and digging (Fig. 1e), and also exhibited a nesting deficit (considered by some to be a home-cage social behavior) compared to littermate controls (Fig. 1f).

We next assessed learning and memory, a behavioral domain significantly impaired in both *Cdkl5* constitutive knockout mice and Nex-cKO mice. Interestingly, Dlx-cKO showed no impairment in spontaneous alternation behavior on the Y-maze, suggesting intact working memory (Fig. 1g, h). On the Barnes maze, an assay of spatial learning and memory, Dlx-cKO did not show a significant difference from wild-type controls in the number of errors made on probe trials (Fig. 1i). However, when the target hole location was altered to the opposite side of the maze after acquisition (reversal probe trials), Dlx-cKO mice demonstrated a significant impairment in adjusting to this new target hole (Fig. 1j). In addition, Dlx-cKO mice show an increased number of perseverations on reversal probe trials, or visits to the previous target hole location (Fig. 1k). These results suggest that Dlx-cKO mice have intact learning and memory, but impaired cognitive flexibility, another feature reminiscent of autism spectrum disorders.

Taken together, Dlx-cKO mice show a set of autistic-like features with relatively preserved learning and memory that stand in contrast to previous findings in Nex-cKO mice, which exclusively show impaired learning and memory[14]. These results demonstrate a segregation of behavioral deficits in *Cdkl5* conditional knockout mice, suggesting that impaired learning and autistic-like features have distinct cellular origins in CDD.

### Dlx-cKO mice show aberrant circuit activation.
In previously established mouse models of ASD, the finding of excitation-inhibition (*E/I*) imbalance has emerged as a prominent theme underlying patterns of circuit and network dysfunction[18,19]. Specifically, hyperexcitability at the circuit and network levels has been found to be associated with many of the autistic-like features reported in these models. Importantly, reversal of hyperexcitability in these models is often sufficient to ameliorate disease

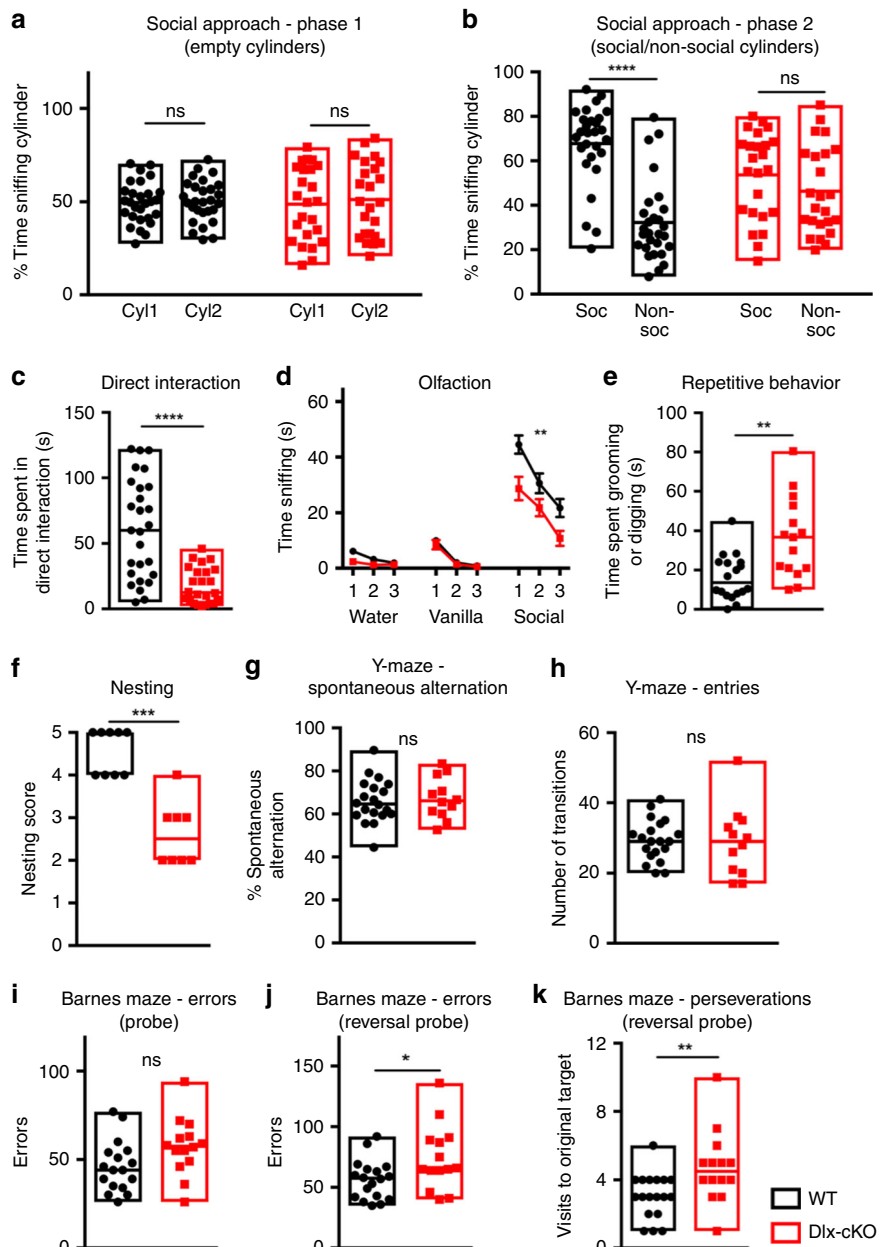

**Fig. 1** Dlx-cKO mice exhibit an autistic-like phenotype. **a** Dlx-cKO mice showed similar times spent interacting with empty cylinders during the habituation phase of the three-chambered social assay. Cyl1/Cyl2: empty cylinders 1 and 2. Soc: social cylinder containing a stimulus mouse. Non-soc: nonsocial cylinder containing an object. (WT, $n = 28$; Dlx-cKO, $n = 26$; paired $t$-test). **b** Wild-type mice showed a significant preference for interacting with the cylinder containing a social stimulus, but Dlx-cKO mice did not show a significant difference in time spent sniffing the social and nonsocial cylinders (paired $t$-test). **c** Dlx-cKO mice also spent significantly reduced time initiating social contact during the direct interaction phase (unpaired $t$-test). **d** Dlx-cKO mice showed unaltered ability to discriminate between odors and habituation to the same odor upon repeated presentation. However, Dlx-cKO mice show a reduced time sniffing a social scent compared to WT (WT, $n = 20$; Dlx-cKO, $n = 14$; two-way ANOVA with Holm-Sidak's multiple comparisons test; error bars demonstrate s.e.m.). **e** Dlx-cKO mice spent significantly more time engaging in repetitive behaviors, including grooming and digging, within a 10-min period in a home-cage like environment (WT, $n = 18$; Dlx-cKO, $n = 16$; unpaired $t$-test). **f** Dlx-cKO mice show significantly reduced nesting scores on the nesting assay (WT, $n = 9$; Dlx-cKO, $n = 8$; Mann–Whitney test). **g** Dlx-cKO mice show unaltered spontaneous alternation percentage in comparison to WT mice (WT, $n = 20$; Dlx-cKO: $n = 12$; unpaired $t$-test). **h** The total number of entries made were not significantly different between Dlx-cKO and WT mice (unpaired $t$-test). **i** Dlx-cKO mice did not show a significant difference in the total number of errors made on forward probe trials of the Barnes maze (WT, $n = 18$; Dlx-cKO: $n = 14$; unpaired $t$-test). **j** Dlx-cKO mice made significantly more errors on the reversal probe trials of the Barnes maze (unpaired $t$-test). **k** On the reversal probe trials, Dlx-cKO mice also made significantly more perseverations, or visits to the original target hole location (Mann–Whitney test). $^{*}p < 0.05$, $^{**}p < 0.01$, $^{***}p < 0.001$, and $^{****}p < 0.0001$. Source data are provided as a Source Data file

phenotypes[20,21], demonstrating a causal role for *E/I* imbalance in generating the features of social impairment and stereotypy.

The exhibition of autistic-like features in Dlx-cKO mice, coupled to the previous findings of *E/I* imbalance in other models of ASD, prompted us to investigate any circuit irregularities in CDD mouse models. We first used voltage-sensitive dye imaging (VSDI), a technique that has been consistently used in mouse models of neurologic disease to assess circuit function with high spatiotemporal resolution[14,22–25]. We conducted our experiments with WT, Dlx-cKO, and Nex-cKO mice, to facilitate a side-by-side comparison of circuit function in the context of the distinct behavioral profiles of Dlx-cKO and Nex-cKO mice.

We began by interrogating the function of the perforant path-dentate gyrus pathway, a hippocampal microcircuit often perturbed in disease states that critically relies on both excitation and inhibition for proper function[26–28]. Using VSDI responses as a readout, we assessed the layer-specific responses of the molecular layer, granule cell layer, as well as the downstream hilar region in response to perforant path stimulation (Fig. 2a). We used a paired-pulse paradigm, which assesses both the baseline excitability of the circuit and its short-term plasticity. We first focused our analysis on the response of the granule cell layer (Fig. 2b–e), which contains the cell bodies of the principal neurons of the dentate gyrus. WT mice showed comparable responses in the granule cell layer to the first and second stimuli without significant facilitation or depression (Fig. 2b–d), showing a paired-pulse ratio that was not significantly different from one (Fig. 2e). Neither Nex-cKO or Dlx-cKO mice showed a significant difference from WT in their peak responses to the first stimulation, suggesting similar baseline excitability of the dentate gyrus (Fig. 2b, c). Interestingly, Nex-cKO mice showed a diminished peak response to the second stimulus, with a paired-pulse ratio significantly less than one (Fig. 2b, d, e). In contrast to Nex-cKO mice, Dlx-cKO mice exhibited paired-pulse facilitation, evident by an increase in depolarization upon a second successive stimulus and a peak ratio significantly greater than one (Fig. 2d, e). Importantly, this aberrant paired-pulse facilitation in Dlx-cKO was also observed in an upstream input region (molecular layer) and downstream output region (hilus) (Fig. 2f–k). Furthermore, we repeated this experiment at a second stimulation intensity and observed similar results (Supplemental Fig. 3). Taken together, our findings reveal that the distinct behavioral phenotypes of *Cdkl5* conditional knockout mice are accompanied by contrasting patterns of circuit dysfunction. Specifically, Dlx-cKO mice show circuit hyperexcitability in the form of aberrant paired-pulse facilitation, whereas Nex-cKO mice show circuit hypo-excitability in the form of aberrant paired-pulse depression.

**Dlx-cKO mice show enhanced excitatory synaptic transmission**. Given the finding of circuit hyperexcitability in Dlx-cKO mice, we next investigated the synaptic origins of this *E/I* imbalance. Previous studies have found enhanced synaptic transmission and long-term potentiation in the hippocampal CA1 region of *Cdkl5* constitutive knockout mice[16]. Interestingly, these changes were not fully recapitulated in the glutamatergic conditional knockout mice, suggesting that CDKL5 function in other neuronal populations, such as GABAergic neurons, may contribute to synaptic hyperexcitability[14]. Therefore, using whole-cell patch-clamp recordings, we assessed both excitatory and inhibitory synaptic transmission in the hippocampal CA1 region in Dlx-cKO mice. We found that Dlx-cKO mice exhibited significantly enhanced frequency of miniature spontaneous excitatory postsynaptic currents (mEPSCs), but no changes in amplitude or kinetics of these events (Fig. 3a–e and Supplemental

Fig. 4A, B). In contrast, miniature spontaneous inhibitory post-synaptic currents (mIPSCs) were not significantly different in either amplitude, frequency, or kinetics between Dlx-cKO and WT mice (Fig. 3f–j and Supplemental Fig. 4C, D). This suggests that despite CDKL5 loss in GABAergic neurons, adult Dlx-cKO mice do not show a significant alteration of inhibitory signaling at CA1 pyramidal neurons. In turn, synaptic transmission between excitatory neurons is enhanced upon selective loss of CDKL5 in GABAergic neurons. Taken together, our results support a shift in *E/I* balance in Dlx-cKO mice toward hyperexcitability that is consistent with our circuit findings.

**Dlx-cKO mice have an increase in postsynaptic NMDA receptors**. The finding of excess glutamatergic synaptic transmission in Dlx-cKO mice suggests a potential alteration in the molecular composition of postsynaptic receptors. Indeed, previous studies have suggested a role for CDKL5 in regulating synaptic receptor composition and homeostasis[16,29]. For example, excessive glutamatergic synaptic transmission can result from an enhancement of postsynaptic AMPA receptors (AMPAR), NMDA receptors (NMDAR), or both. We therefore examined the composition of major ionotropic glutamate receptors in post-synaptic density membrane preparations from Dlx-cKO and WT mice, including GluA1, GluA2, GluN1, GluN2A, and GluN2B. Interestingly, we found a selective increase in the protein level of NMDARs, but not AMPARs in Dlx-cKO mice (Fig. 4). Specifically, the NMDAR subunits GluN1 and GluN2B were significantly elevated in Dlx-cKO mice. (Fig. 4d, f). These findings are consistent with results from *Cdkl5* constitutive knockout mice[16] and suggest a non-cell autonomous mechanism underlying the enhancement of glutamatergic synaptic transmission through CDKL5 loss from GABAergic neurons.

**NMDAR blockade ameliorates autistic-like features in Dlx-cKO**. Our findings in Dlx-cKO mice raised the possibility that enhanced NMDAR signaling may underlie the autistic-like phenotype exhibited by these mice. We first investigated the extent to which NMDAR signaling is responsible for the repetitive behavior phenotype in Dlx-cKO mice. We used low doses of memantine, a non-competitive NMDA receptor blocker that has been previously found to rescue behavioral deficits in numerous mouse models of neurodevelopmental disorders[21,30,31]. We observed that acute memantine administration (5 mg kg$^{-1}$, intraperitoneal) ameliorated excessive grooming in Dlx-cKO mice, significantly reducing both total time spent grooming and the duration of the longest grooming bout (Fig. 5a, b). Next, we employed a modified dyadic interaction assay to quantitatively assess social interactions in drug- and vehicle-treated Dlx-cKO mice. Whereas vehicle-administered Dlx-cKO mice showed significantly reduced social interaction, memantine ameliorated this deficit, significantly increasing the time spent by Dlx-cKO mice in social interaction (Fig. 5c). This effect was specific to Dlx-cKO mice, as memantine normalized the ratio of time spent in social interaction in pairs of Dlx-cKO and WT control mice co-tested side-by-side in identical cages (Fig. 5d). Taken together, these results suggest that enhanced NMDAR signaling underlies, at least in part, the autistic-like features of Dlx-cKO mice, and that an acute reduction of NMDAR activity can ameliorate these phenotypes.

**NMDAR blockade improves autistic-like features in CDKL5 R59X**. To investigate the translational potential of our findings in Dlx-cKO mice, we generated a novel mouse model of CDD, *Cdkl5$^{R59X}$* (referred to as R59X). R59X mice bear a knock-in

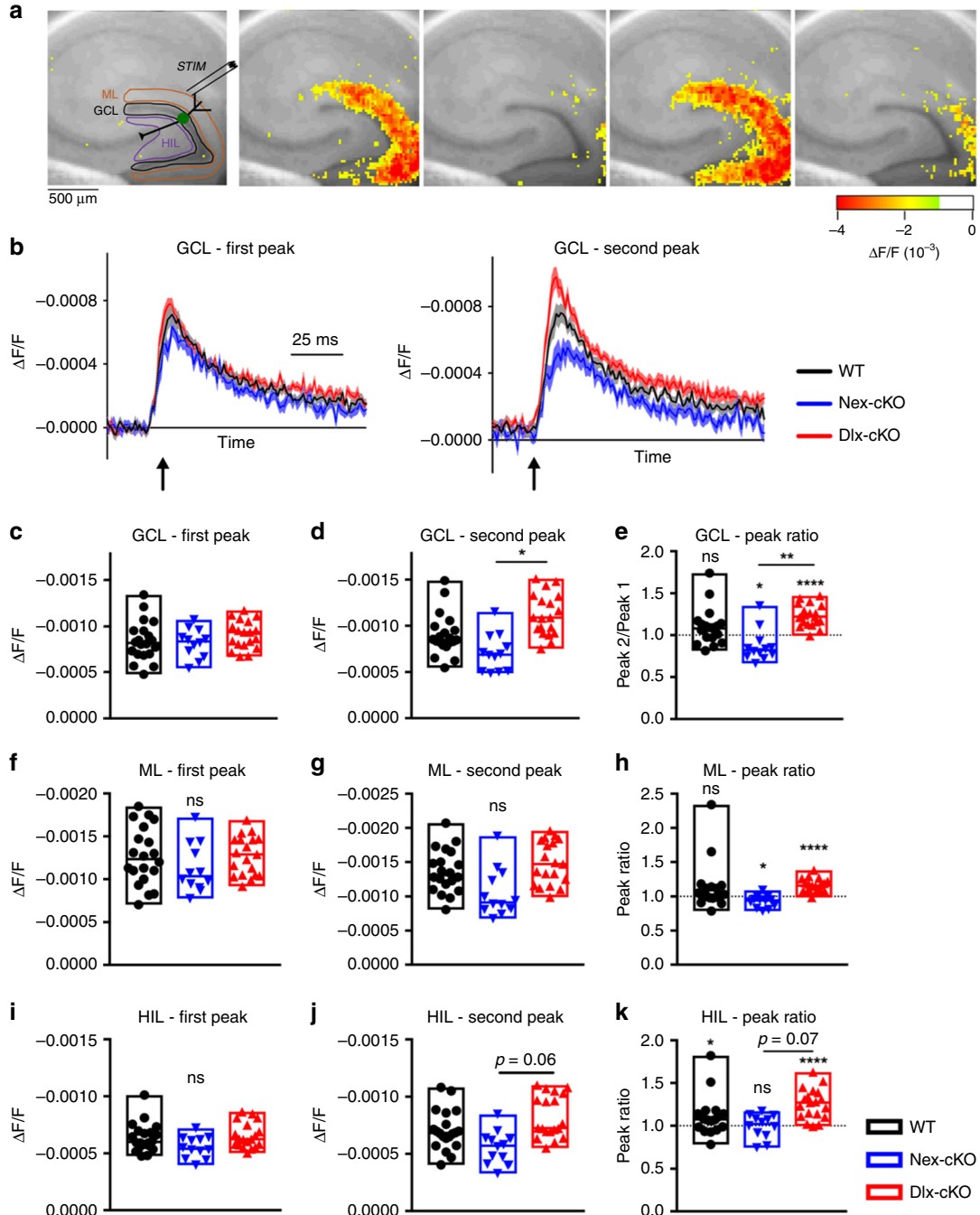

**Fig. 2** Dlx-cKO mice show aberrant paired-pulse facilitation. **a** Schematic of voltage-sensitive dye imaging (VSDI) showing the site of stimulation at the perforant path and regions of interest used for quantitative analysis (GCL, granule cell layer; ML, molecular layer; HIL, hilus). Snapshots of VSDI responses in a wild-type slice, from left to right: baseline, peak response to first stimulus, baseline immediately prior to second stimulus, peak response to second stimulus, 100 ms after second peak response (200 μA x 2 stimuli, with 200 ms inter-stimulus interval). **b** Averaged VSDI responses ($\Delta F/F$) in the GCL for WT, Nex-cKO, and Dlx-cKO mice in response to each of the two stimuli. Depolarization (decrease in $\Delta F/F$) is displayed as an upward signal, while hyperpolarization (increase in $\Delta F/F$) is displayed as a downward signal. Error envelopes represent mean $+/-$ s.e.m. **c** Peak amplitudes of the GCL response to the first stimulus were not different between WT, Nex-cKO, and Dlx-cKO mice (linear mixed effect analysis). **d** Peak amplitudes of the GCL response to the second stimulus were significantly different between Nex-cKO and Dlx-cKO mice (linear mixed effect analysis with Tukey's correction for multiple comparisons). **e** Ratios of peak responses in the GCL. Nex-cKO slices showed significant paired-pulse depression, whereas Dlx-cKO slices showed significant paired-pulse facilitation (one-sample $t$-tests for each of WT, Nex-cKO, and Dlx-cKO, two-tailed). Dlx-cKO also showed significantly increased paired-pulse ratio in comparison to Nex-cKO (linear mixed effect analysis with Tukey's correction for multiple comparisons). **f–k** Averaged VSDI responses ($\Delta F/F$) in the ML and HIL for WT, Nex-cKO, and Dlx-cKO mice in response to each of the two stimuli (linear mixed effect analysis). **h** Peak ratios in ML (one-sample $t$-test or Wilcoxon signed-rank test, two-tailed). **k** Peak ratios in HIL (one-sample $t$-test or Wilcoxon signed-rank test, two-tailed). For all experiments, $n = 20$ slices/5 mice for WT, $n = 12$ slices/4 mice for Nex-cKO, and $n = 19$ slices/5 mice for Dlx-cKO. *$p < 0.05$, ****$p < 0.0001$. Source data are provided as a Source Data file

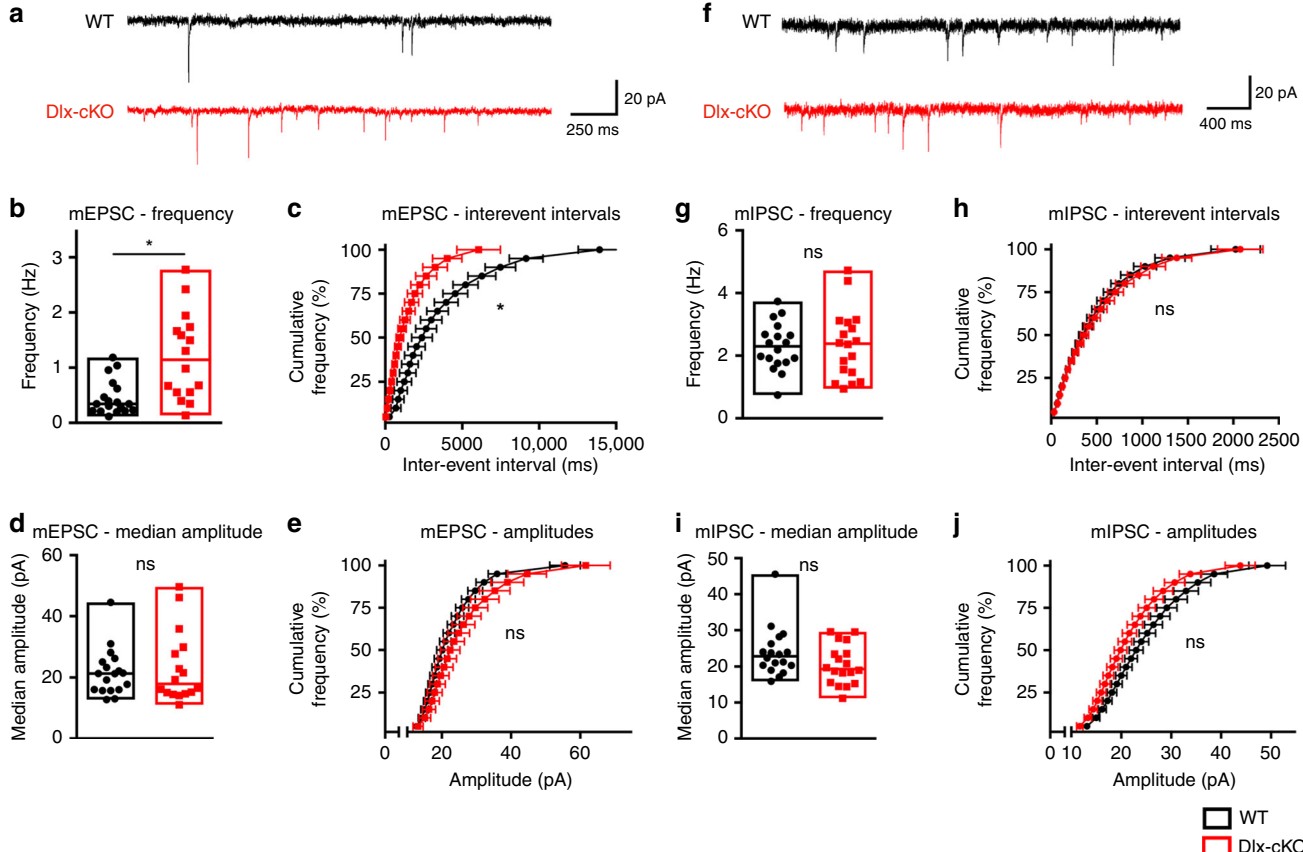

**Fig. 3** Enhanced excitatory synaptic transmission but unaltered inhibitory synaptic transmission in Dlx-cKO mice. **a** Representative mEPSC traces for cells from WT and Dlx-cKO mice. **b** Dlx-cKO mice show significantly enhanced mEPSC frequency at CA1 pyramidal neurons (linear mixed effect analysis). **c** The cumulative frequency distribution of mEPSC inter-event intervals is also significantly shifted in Dlx-cKO mice (linear mixed effect analysis with least-square means post-hoc testing). **d** The mEPSC median amplitude is unaltered in Dlx-cKO mice (linear mixed effect analysis). **e** The cumulative frequency distributions of mEPSC amplitudes are not significantly different between Dlx-cKO and WT mice (linear mixed effect analysis). **f** Representative mIPSC traces for cells from WT and Dlx-cKO mice. **g** Dlx-cKO mice show unaltered mIPSC frequency at CA1 pyramidal neurons (linear mixed effect analysis). **h** The cumulative frequency distributions of mIPSC inter-event intervals were not significantly different between Dlx-cKO and WT mice (linear mixed effect analysis). **i** The median amplitude of mIPSCs is not significantly different between Dlx-cKO and WT mice (linear mixed effect analysis). **j** The cumulative frequency distributions of mIPSC amplitudes are not significantly different between Dlx-cKO and WT mice (linear mixed effect analysis). For mEPSCs: WT, $n = 18$ cells/3 mice; Dlx-cKO, $n = 16$ cells/3 mice; for mIPSCs: WT, $n = 18$ cells/3 mice; Dlx-cKO, $n = 18$ cells/3 mice. *$p < 0.05$. Error bars s.e.m. Source data are provided as a Source Data file

mutation mimicking a human CDD nonsense mutation at arginine 59, leading to the loss of CDKL5 kinase function (Fig. 6a). We found that full-length CDKL5 protein expression was abolished in R59X mice, similar to a knockout model we previously developed[4] (Supplemental Fig. 5). Similar to Dlx-cKO mice and other mouse models of CDD, R59X mice demonstrate autistic-like features, including increased repetitive behaviors and diminished social interaction (Fig. 7).

We next investigated the extent to which enhanced NMDAR signaling is also present in R59X mice. Using a similar approach as for Dlx-cKO mice, we examined the protein levels of NMDAR and AMPAR subunits in postsynaptic density fractions (Fig. 6b–g). In R59X mice, we found an upregulation of NMDAR subunit levels, with a selective increase in GluN2B, mirroring previous findings in *Cdkl5* constitutive knockout mice[16] (Fig. 6g).

Using R59X mice as a model of CDD, we next examined the extent to which enhanced NMDA signaling is responsible for CDD-related behavioral deficits. We first tested the possibility that enhanced NMDAR signaling may underlie impaired learning and memory in CDKL5-deficient mice. Using the Y-maze assay, we found that R59X mice demonstrated a robust working

memory deficit, demonstrated by reduced spontaneous alternation behavior, similar to our previous findings in both *Cdkl5* constitutive KO and Nex-cKO mice[14] (Fig. 7a). Notably, acute memantine administration at a range of doses (1.25–10 mg kg$^{-1}$, intraperitoneal) did not ameliorate this behavioral deficit, suggesting that excess NMDAR signaling is unlikely to be the major mechanism underlying learning and memory deficits in R59X mice (Fig. 7a).

In contrast, we observed that acute memantine administration (5 mg kg$^{-1}$, intraperitoneal) rescued excessive stereotypic behaviors in R59X mice, significantly reducing both total time spent grooming and the duration of the longest grooming bout to a level similar to that of WT mice (Fig. 7b, c). Similarly, whereas vehicle-treated R59X mice showed significantly reduced social interaction, memantine ameliorated this deficit, significantly increasing the time spent by R59X mice in social interaction (Fig. 7d). This effect was specific to R59X mice, as memantine normalized the ratio of time spent in social interaction in pairs of R59X and WT control mice co-tested side-by-side in identical cages (Fig. 7e). Taken together, these results suggest that enhanced NMDAR signaling underlies the autistic-like features,

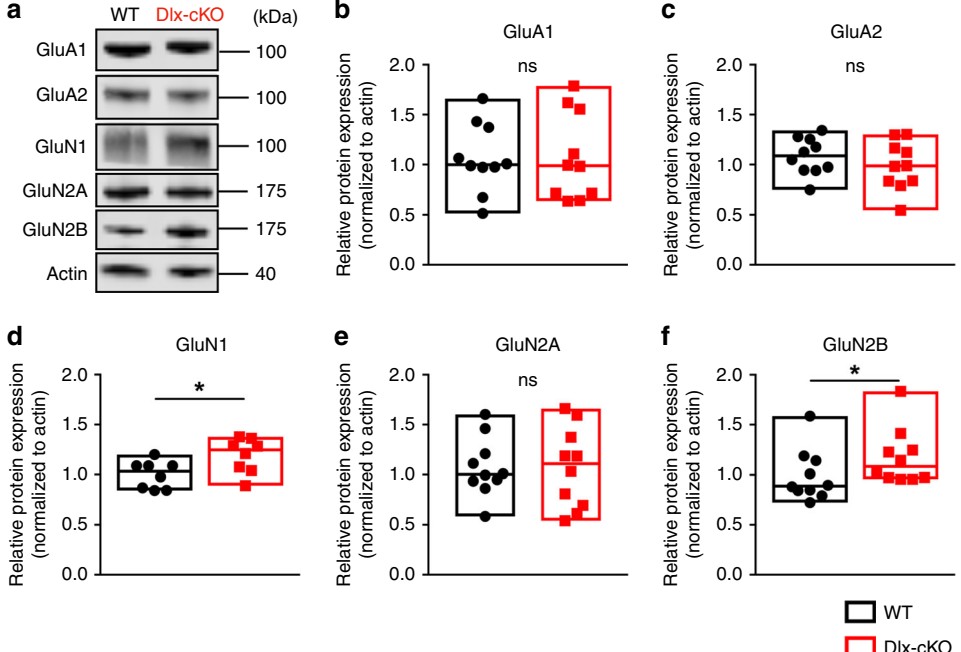

**Fig. 4** Dlx-cKO mice have an upregulation of NMDA receptors at the postsynaptic membrane. **a** Representative western blot results for several major ionotropic glutamate receptor subunits from postsynaptic density membrane fractions. Samples are from pairs of WT and Dlx-cKO littermates. **b**–**f** Dlx-cKO mice show a selective increase in levels of GluN1 and GluN2B, two of the major subunits that compose the NMDA receptor (WT, $n = 10$ mice; Dlx-cKO, $n = 10$ mice; unpaired $t$-test or Mann–Whitney test). Full-scan western blots of all samples are available in Supplemental Fig. 6. *$p < 0.05$. Source data are provided as a Source Data file

but not learning and memory deficits, in CDKL5-deficient mice, and reveal the translational potential of this pathway in the treatment of CDD.

## Discussion

Our behavioral studies in Dlx-cKO mice revealed an autistic-like phenotype characterized by reduced social preference and increased stereotypic behaviors, but relative preservation of other behaviors such as learning and memory, motor coordination, and anxiety-related behaviors. This behavioral profile stands in contrast to our previous findings in Nex-cKO mice, which exhibit selectively impaired learning and memory. Our results delineate the distinct cellular origins of CDD-related behavioral phenotypes: whereas impaired learning primarily originates from the loss of CDKL5 in forebrain glutamatergic neurons, autistic-like features primarily originate from the loss of CDKL5 in forebrain GABAergic neurons. Interestingly, anxiety-related behavior, locomotion, and motor coordination were not altered in either Nex-cKO or Dlx-cKO mice. These additional phenotypes may be mediated by 1) CDKL5 function in other cell populations and brain regions and/or 2) the synergistic functions of CDKL5 in forebrain glutamatergic and GABAergic neurons.

Our findings in *Cdkl5* conditional knockout mice support a model whereby CDKL5 loss in glutamatergic and GABAergic neurons lead to divergent changes in *E/I* balance that generate distinct behavioral deficits. We show that in Dlx-cKO mice, hippocampal circuit hyperexcitability is coupled with autistic-like features, a finding corroborated in numerous other mouse models of autism[20,21,32,33]. In specific cases, autistic-like behaviors have been linked to a dysfunction of inhibitory neuronal transmission[34–36]. In contrast, we show that in Nex-cKO mice, hippocampal circuit hypo-excitability is linked to learning and memory deficits. Interestingly, reduced *E/I* ratio has been demonstrated in numerous mouse models of intellectual disability[37,38]. Therefore,

our studies reveal a novel avenue toward treating specific CDD-related symptoms, but also suggest that modulation of *E/I* balance in CDD may be a double-edged sword. To date, the majority of reported CDD patients have been predicted to carry loss-of-function mutations, such as nonsense, indels, or missense mutations abolishing CDKL5 kinase activity[39], suggesting that our knockout and R59X knockin mouse models may carry a high translational relevance. In order to develop targeted therapies for CDD, additional studies are needed to elucidate the specific signaling pathways and neural circuits responsible for each of the behavioral symptoms.

In our VSDI studies of Nex- and Dlx-cKO mice, we employed a paired-pulse stimulation paradigm, which assesses the integration of excitatory and inhibitory synaptic transmission in the dentate gyrus. Interestingly, previous work has shown that NMDA signaling is a necessary component for the paired-pulse facilitation response in the dentate gyrus, suggesting a potential mechanism for circuit excitability in Dlx-cKO mice[40]. On the other hand, the aberrant paired-pulse depression in Nex-cKO mice may result from enhanced inhibition[14], given that feedforward and feedback inhibition play key roles in regulating the activity of the dentate gyrus[41,42]. In our electrophysiologic studies of Dlx-cKO mice, the most prominent change was, surprisingly, an enhancement in glutamatergic synaptic transmission. This change likely arises from a non-cell autonomous mechanism, especially in the absence of a concomitant change in GABAergic-glutamatergic inhibitory synaptic transmission. For example, the loss of CDKL5 in GABAergic interneurons may lead to a modulation of presynaptic release probability at glutamatergic synapses. Alternatively, enhanced excitatory synaptic transmission may be a homeostatic response to a transient alteration of GABAergic signaling during early development. Supporting this notion, a previous study has found that GABAergic transmission cooperates with NMDA receptor activation during development for excitatory synapse formation[43]. Furthermore, the loss of

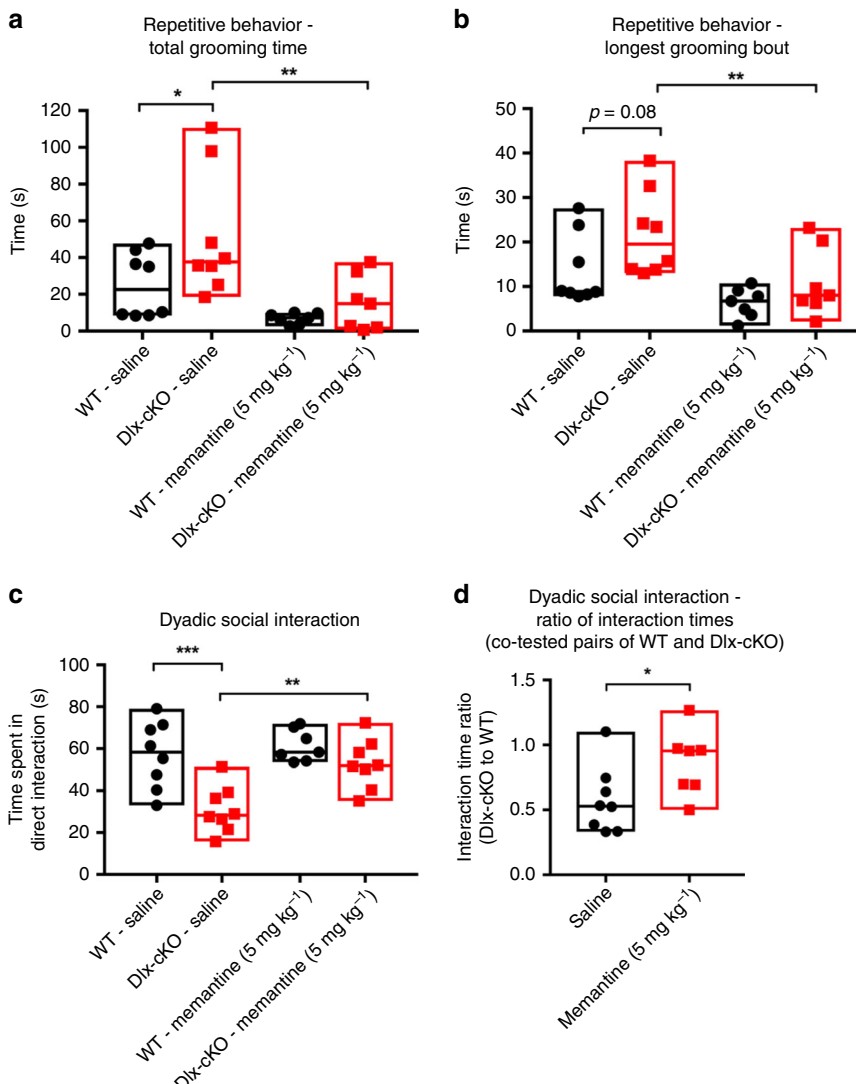

**Fig. 5** Acute NMDAR blockade ameliorates autistic-like features in Dlx-cKO mice. **a** Total grooming in a home-cage like environment is significantly increased in saline-treated Dlx-cKO mice in comparison to WT, and memantine at 5 mg kg$^{-1}$ significantly rescued the increased grooming phenotype in Dlx-cKO mice ($n = 8$ for saline groups; $n = 7$ for memantine groups; one-way ANOVA with Holm-Sidak's multiple comparisons test). **b** Memantine significantly reduces the duration of the longest grooming bout in Dlx-cKO mice. **c** On the dyadic social assay, saline-treated Dlx-cKO mice spend significantly less time initiating social interaction with a novel stimulus mouse in comparison to WT. Memantine at 5 mg kg$^{-1}$ resulted in a significant increase in time spent in social interaction ($n = 8$ for saline groups; $n = 7$ for WT—memantine; $n = 8$ for Dlx-cKO—memantine; one-way ANOVA with Holm-Sidak's multiple comparisons test). **d** For co-tested pairs of Dlx-cKO and WT mice, the ratio of time spent in direct interaction was significantly reduced in saline-treated mice. Memantine normalized this ratio, indicating a differential effect of this drug on social interaction in WT and Dlx-cKO mice ($n = 7$ for all groups; unpaired $t$-test). *$p < 0.05$, **$p < 0.01$, and ***$p < 0.001$. Source data are provided as a Source Data file

CDKL5 may lead to different disruptions in distinct populations of forebrain interneurons that as of yet we are unable to dissect from our Dlx5/6-Cre-driven dissection. Future studies looking into the role of CDKL5 at specific sub-populations of interneurons will provide clarity on its non-cell autonomous roles.

Interestingly, despite the findings of synaptic and circuit-level hyperexcitability, we did not observe spontaneous seizure-like behaviors in Dlx-cKO mice during our routine assays, at least in male mice and at the ages examined. This is similar to what has been observed in constitutive loss-of-function models of CDD where EEG recordings have not revealed spontaneous seizures[4,5,16]. However, this does not exclude the possibility that seizure susceptibility is altered in the absence of CDKL5 under certain environmental conditions and during specific periods in development and aging. Thus far, the lack of a robust seizure phenotype in CDD mouse models suggests that neural network

differences between human and mouse brains may prevent the development of overt seizures in CDKL5-deficient mice[44]. In addition, given the contrasting profiles of hypo- and hyperexcitability in Nex-cKO and Dlx-cKO mice, respectively, the state of circuit excitability in constitutive loss-of-function models of CDD remains to be further characterized. Indeed, previous studies have revealed changes in synapse density, distribution, and plasticity of both excitatory and inhibitory neurons[15,16]. Therefore, it is possible that both pro- and anti-epileptogenic changes occur in the CDKL5-deficient brain, each playing a distinct role in mediating the various behavioral deficits of CDD.

The finding of enhanced NMDAR signaling reveals a novel synaptic mechanism that may potentially underlie the behavioral deficits of mouse models of CDD. Our results in Dlx-cKO mice suggest that GABAergic dysfunction leads to an enhancement of NMDAR signaling, which in turn is responsible for autistic-like

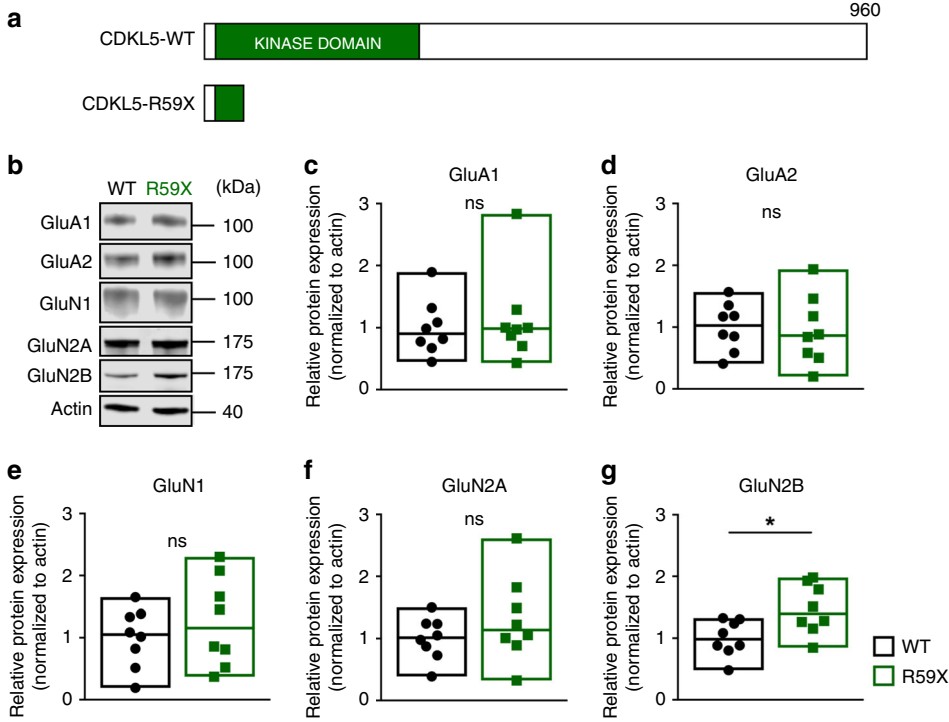

**Fig. 6** R59X mice have an upregulation of NMDA receptors at the postsynaptic membrane. **a** Schematic of the R59X nonsense mutation at arginine 59, which results in early truncation of the catalytic domain of CDKL5. **b** Representative western blot results for several major ionotropic glutamate receptor subunits from postsynaptic density membrane fractions. Samples are from pairs of WT and R59X littermates. **c–g** R59X mice show a selective increase in level of GluN2B, a major subunit of the NMDA receptor (WT, $n = 8$ mice; R59X, $n = 8$ mice; unpaired $t$-test or Mann–Whitney test). Full-scan western blots of all samples are available in Supplemental Fig. 6. Source data are provided as a Source Data file

features. We validate this mechanism in R59X mice, demonstrating that the acute reduction of NMDAR signaling ameliorates select CDD-related behavioral deficits. Whereas acute memantine administration specifically rescued repetitive behavior and significantly improved social interaction in R59X mice, it did not positively affect the learning and memory deficit in R59X mice. Altogether with our previous findings in Nex-cKO mice[14], our studies suggest that distinct subsets of CDD-related phenotypes are mediated by distinct synaptic and circuit mechanisms. Future studies are expected to refine these mechanisms by testing brain region- and circuit-specific mechanisms for specific CDD-related phenotypes. For example, given that we found enhanced NMDAR subunit levels in our postsynaptic density membrane fractions, we cannot rule out the possibility that extra-synaptic NMDAR levels are also altered in Dlx-cKO and R59X mice. Indeed, memantine is known to act at both synaptic and extra-synaptic receptors in a dose-dependent manner, and its effects on behavior may be mediated by both mechanisms[45]. Although it may be difficult to distinguish between these two populations of NMDARs in vivo, further work in this direction can elucidate the mechanism that underlies the enhancement of synaptic NMDAR signaling in mouse models of CDD.

Taken together, our studies delineate the forebrain GABAergic origins of autistic-like features in CDKL5-deficient mice, showing for the first time that CDKL5 is required in this diverse population of neurons for proper neural development and function. At the synaptic level, we reveal an enhancement of NMDAR signaling that contributes to circuit hyperexcitability. Finally, in a novel disease model of CDD, we show that acute reduction of NMDAR signaling can ameliorate behavioral deficits, highlighting a potentially important mechanism for CDD-related phenotypes in mice and supporting a novel therapeutic avenue for the treatment of these symptoms in CDD patients.

## Methods

**Mouse strains.** The *Cdkl5* floxed mouse line with Cre-dependent excision of exon 7 (based on revised *Cdkl5* gene nomenclature; formerly exon 6) was used for the generation of conditional knockout mice[14]. Dlx5/6-Cre, a mouse line expressing Cre in forebrain inhibitory neurons[17], was obtained from Jackson Laboratories and maintained in the C59BL/6J background (Stock No. 008199). The CDKL5 R59X knock-in line was generated as follows: a targeting vector was designed to insert a *frt*-flanked neomycin resistance (neo) cassette downstream of exon 5, and a single-nucleotide change of C to T, leading to a nonsense mutation at arginine 59 (R59X) of the *Cdkl5* gene. The construct was electroporated into C57BL/6N embryonic stem (ES) cells. Correctly targeted ES cells were injected into BALB/c blastocysts and resulting chimeric mice were bred with B6.Cg-Tg(ACTFLPe)9205Dym/J (Jackson Laboratories, Stock No. 005703) to remove the neo cassette. Resulting offspring were bred to C57BL/6J mice (Jackson Laboratories, Stock No. 000664) for at least ten generations. The R59X knock-in mouse line has been deposited at Jackson Laboratories (Stock No. 028856).

**Mouse husbandry.** Experiments were conducted in accordance with the ethical guidelines of the National Institutes of Health and with the approval of the Institutional Animal Care and Use Committee of the University of Pennsylvania. *Cdkl5flox* and R59X mice were backcrossed to C57BL/6 for at least ten generations before breeding for experiments. Mouse lines were genotyped using a PCR-based strategy, with their respective primer sequences and other details found on the Jackson Laboratories website under the appropriate stock numbers listed above. Mice were group-housed in cages of three to five in a 12-h light/dark cycle with food and water provided ad libitum.

For breeding of Dlx-cKO and wild-type control mice, each breeding cage consisted of two homozygous female mice (genotype: *Cdkl5flox/Cdkl5flox*) and one male mouse (genotype: Dlx5/6-Cre/+). Male littermates (genotypes: *Cdkl5flox*/y; +/+ and *Cdkl5flox*/y; Dlx5/6-Cre) were weaned at 3 weeks of age and housed together, with all experiments performed on age-matched adult mice between 9 and 12 weeks of age. For behavioral experiments, wild-type control mice (genotype: *Cdkl5flox*/y; +/+) from Nex-cKO breedings[14] were pooled with wild-type control mice from Dlx-cKO breedings (genotype: *Cdkl5flox*/y; +/+). Nex-cKO mice were generated using a similar strategy as described previously[14]. For breeding of R59X and wild-type control mice, each breeding cage consistent of two heterozygous female mice (genotype: R59X/+) and one male mouse (genotype: +/y). Male littermates (genotypes: R59X/y and +/y) were weaned at 3 weeks of age and housed together, with all experiments performed on age-matched adult mice between 9 and 12 weeks of age.

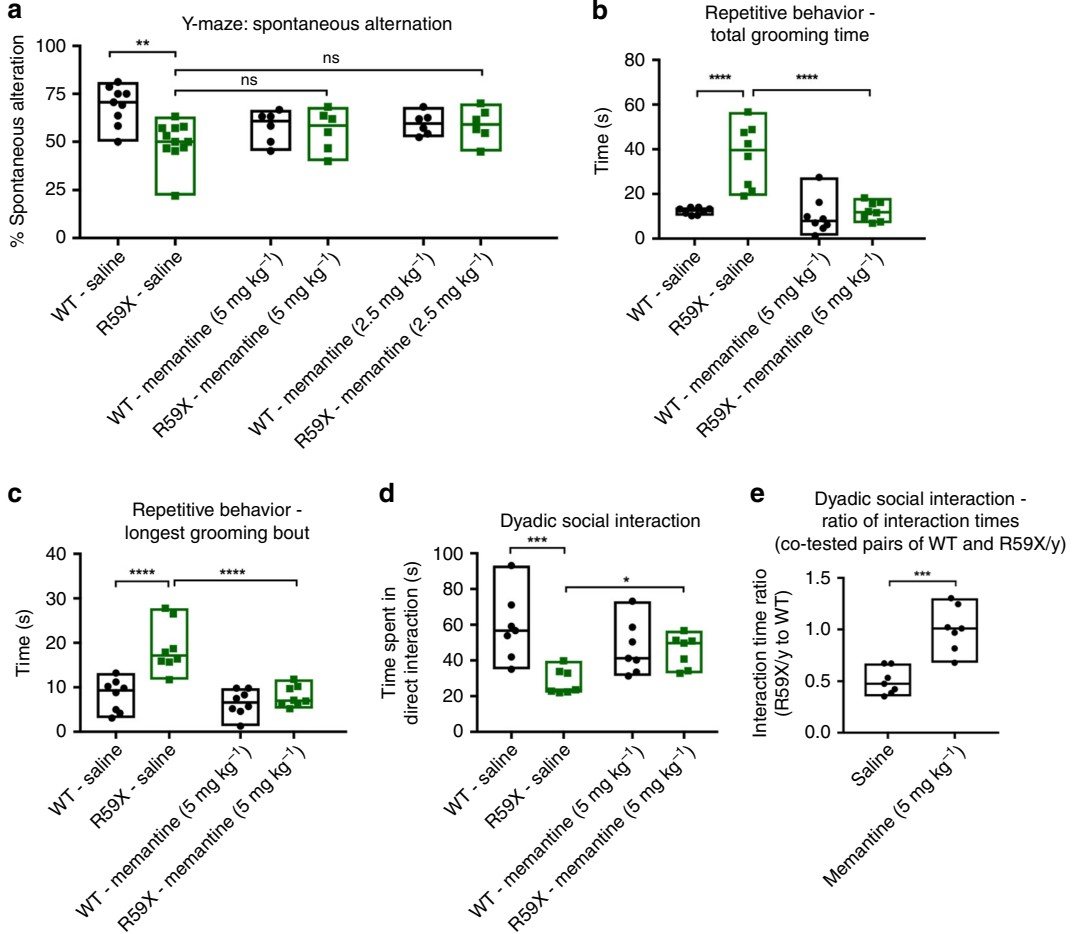

**Fig. 7** Acute NMDAR blockade ameliorates autistic-like features, but not learning and memory, in CDKL5-deficient mice. **a** Spontaneous alternation percentage is significantly reduced in saline-treated R59X mice in comparison to WT mice. However, acute memantine administration at either 2.5 or 5 mg kg$^{-1}$ doses did not significantly rescue the spontaneous alternation deficit (WT-saline, $n = 9$; R59X-saline, $n = 11$; WT-memantine (5 mg kg$^{-1}$), $n = 6$; R59X-memantine (5 mg kg$^{-1}$), $n = 6$; WT-memantine (2.5 mg kg$^{-1}$), $n = 6$; R59X-memantine (2.5 mg kg$^{-1}$), $n = 6$; Kruskal–Wallis test with Dunn's multiple comparisons test). **b** Total grooming in a home-cage like environment is significantly increased in saline-treated R59X mice in comparison to WT, and memantine at 5 mg kg$^{-1}$ significantly ameliorated the increased grooming phenotype in R59X mice ($n = 8$ for all groups; one-way ANOVA with Holm-Sidak's multiple comparisons test). **c** The duration of the longest grooming bout is significantly increased in saline-treated R59X mice in comparison to WT, and memantine also ameliorated this aspect of repetitive behavior ($n = 8$ for all groups; one-way ANOVA with Holm-Sidak's multiple comparisons test). **d** On the dyadic social assay, saline-treated R59X mice spend significantly less time initiating social interaction with a novel stimulus mouse in comparison to WT. Memantine at 5 mg kg$^{-1}$ resulted in a significant increase in time spent in social interaction ($n = 7$ for all groups; one-way ANOVA with Holm-Sidak's multiple comparisons test). **e** For co-tested pairs of R59X and WT mice, the ratio of time spent in direct interaction was significantly reduced in saline-treated mice. Memantine normalized this ratio, indicating a differential effect of this drug on social interaction in WT and R59X mice ($n = 7$ for all groups; unpaired $t$-test). $*p < 0.05$, $**p < 0.01$, $***p < 0.001$, and $****p < 0.0001$. Source data are provided as a Source Data fil

**Behavioral assays**. All animal behavioral studies were carried out blinded to genotype. Mice were allowed to habituate to the testing room for at least 1 h before the test, and testing was performed at the same time of day. All animal behaviors were performed on adult male mice at 9–12 weeks of age, and the analysis of behavioral data was carried out by a researcher blinded to genotype.

**Elevated zero maze**. The elevated zero maze (Stoelting, Illinois, USA) consists of a circular-shaped platform elevated above the floor. Two opposite quadrants of the maze are enclosed (wall height, 12 inches), whereas the other two are open (wall height, 0.5 inches). Mice were placed in one of the closed quadrants and their movement traced over the course of 5 min. Analysis, including the quantification of percent of time spent in open arms and the number of entries, was performed manually using a stopwatch. An entry was defined as a transition from a closed to open arm, or vice versa, that involves all four paws.

**Three-chambered social approach assay**. The social choice test was carried out in a three-chambered apparatus that consisted of a center chamber and two end chambers[46,47]. Before the start of the test and in a counter-balanced sequence, one end chamber was designated the social chamber, into which a stimulus mouse would be introduced, and the other end chamber was designed the nonsocial

chamber. Two identical, clear Plexiglas cylinders with multiple holes to allow for air exchange were placed in each end chamber. In the habituation phase of the test, the test mouse was placed in the center chamber and allowed to explore all three chambers for 10 min. During this acclimation period, baseline measurements of how much time the mouse spent in each of the three chambers and the distance traveled by the test mouse were collected.

In the social choice phase of the test, a stimulus mouse (adult gonadectomized A/J mice; The Jackson Laboratory) was placed in the cylinder in the social chamber while a novel object was simultaneously placed into the other cylinder in the nonsocial chamber. During the 5-min social choice period, chamber times and numbers of transitions among chambers were again recorded.

In the direct social interaction test, the cylinders were removed simultaneously following the social choice test, and the amount of time test and stimulus mice spent in direct contact (sniffing, allogrooming) was measured. If fighting persisted for more than several seconds, the mice were removed from the apparatus and excluded from the study.

**Y-maze**. Spontaneous alternation behavior was measured on a Y-maze apparatus (San Diego Instruments, California, USA), composed of three arms (Arm A: 8in. x 5in. x 3in.; Arms B and C: 6in. x 5in. x 3in.). For habituation, the test mouse was

placed in each of the three arms, facing the center, and allowed to make one choice to enter another arm. For testing, the mouse was placed in Arm C, facing the center, and allowed to freely explore the maze for 5 min. A spontaneous alternation was defined an entry into the arm less recently explored. Percent spontaneous alternation was calculated as the number of spontaneous alternations over the total number of entries. For example, the sequence C,B,A,B,C,B,A,C (starting in arm C) resulted in a percent spontaneous alternation of 4/6 = 67%.

**Barnes maze**. Hippocampal-dependent memory was assessed on a Barnes Maze apparatus (San Diego Instruments, California, USA), a circular platform with 36-inch diameter and 20 equally spaced escape holes along the perimeter, one of which leads to a "target" escape box. Bright lighting was used as stimulus to complete the task. The assay consisted of five phases: adaptation, forward acquisition training, forward probe trials, reversal training, and reversal probe trials. For adaptation, each mouse was placed in a dark start chamber in the middle of the maze for 10 s, then uncovered and guided gently to the escape box. Forward acquisition training consisted of two trials per day for 4 days, with each mouse starting in the dark start chamber in the middle of the maze and subsequently allowed to explore the maze for 3 min. The trial ends when the mouse enters the target escape hole or after 3 min have elapsed, after which the mouse is guided gently to the escape hole. After reaching the escape hole, the mouse is allowed to remain there for 1 min. Forward probe trials were conducted on day 5, 24 h after the last training day. During the probe trial, the maze is in the same position as the training days, and the target hole is closed. Each trial lasted 90 s, during which the number of errors (pokes into non-target holes) made prior to reaching the target hole is quantified. Days 6–10 consisted of reversal training, conducted using a similar protocol as forward acquisition training, except that the target was a stable escape hole moved 180° from its location during forward acquisition training. Reversal probe trials were conducted on day 11 and identically to forward probe trials. In addition to quantifying the number of total errors, the number of perseverations, defined as pokes into the previous target hole during forward acquisition training, was counted.

**Locomotor assay**. Locomotor activity was measured by beam breaks in a photo-beam frame (Med Associates, Vermont, USA). Mice were individually placed into a clean home-cage like environment lined with bedding and resting within a photobeam frame. The number of beam breaks as a measure of locomotor activity was quantified over 30 min in 5-min bins.

**Accelerating Rotarod assay**. Mice were placed on an accelerating Rotarod apparatus (Med Associates) for 16 trials (four trials a day for 4 consecutive days) with at least 15 min of rest between the trials. Each trial lasted for a maximum of 5 min, during which the rod accelerated linearly from 4 to 40 rpm. The amount of time for each mouse to fall from the rod was recorded for each trial.

**Olfaction**. Mice were tested for whether they could detect and differentiate odors in a habituation–dishabituation protocol modified from Yang and Crawley. Mice were presented with cotton-tipped wooden applicators dipped in water, vanilla, or swiped across the bottom of an unfamiliar social cage. Each stimulus was presented for 2 min with a 1-min inter-trial interval. Time spent sniffing was defined as when the animal was oriented with its nose 2 cm or closer toward the cotton tip.

**Repetitive behavior**. Mice were individually placed into a clean home-cage like environment lined with bedding. After allowing 5 min for habituation, 10 min of activity was videotaped for each mouse. The duration of repetitive behavior, defined as grooming or digging, was scored manually using a stopwatch.

**Nesting**. Nesting behavior was scored in accordance with the standard protocol and rating scale originally described by Deacon[48]. Four- to five-week old mice were assessed for amount of cotton material used after 20 h and for the height and shape of the nest. Rating scale as follows: (1) nestlet not noticeably touched (>90% intact); (2) nestlet partially torn (50–90% remaining intact); (3) nestlet mostly shredded but often no identifiable nest site: <50% of the Nestlet remains intact, but <90% is within a quarter of the cage floor area; (4) an identifiable but flat nest: >90% of the Nestlet is torn and the material is gathered into a nest within a quarter of the cage floor area, but the nest is flat, with walls higher than mouse body height (of a mouse curled up on its side) for <50% of its circumference; (5) a (near) perfect nest: >90% of the Nestlet is torn and the nest is a crater, with walls higher than mouse body height for >50% of its circumference.

**Dyadic interaction**. The dyadic interaction test is an abbreviated version of the three-chambered assay to quantitatively assess the duration of direct social interaction in two freely behaving mice. One week prior to the assay, male R59X and wild-type control mice were singly housed to increase the motivation for social interaction and reduce aggression. Age-matched, novel, wild-type male stimulus mice were group-housed. On the day of the test, test and stimulus mice were acclimated to the room for 1 h prior to the start of the assay. Following this period, each test mouse is placed individually in a clear, round Plexiglass cage (Pinnacle

Technology, Lawrence, Kansas, USA) measuring 8 inches tall and 9.5 inches in diameter, with a thin layer of bedding. Following 30 min of habituation in the cage, a novel, age-matched stimulus mouse was placed into the cage, and the mice were allowed to freely interact under video monitoring for 10 min. The total duration of social interaction initiated by the test mouse in the first 2 min was scored, and this included oro-genital sniffing, oro-facial sniffing, close following, placing one or both paws on the stimulus mouse, or allogrooming of the stimulus mouse. "Co-tested pairs" refer to side-by-side tests involving two littermate R59X and wild-type control mice, each interacting with a novel stimulus mouse in a separate cage.

**Drug administration**. Mice were acclimated to daily intraperitoneal (i.p.) injections of vehicle (saline) for 3 days prior to each set of behavioral assays. For each behavioral cohort, the entire cohort was administered either vehicle or drug, with the experimenter blinded to the genotypes of mice throughout. All animal behaviors were performed on adult male mice at 9–12 weeks of age, and the analysis of behavioral data was carried out by a researcher blinded to genotype. Vehicle and drug (memantine hydrochloride, Tocris Biosciences, United Kingdom) were diluted in saline at 10 μL per gram mouse body weight for administration. Vehicle or drug was administered 1 h prior to the start of each behavioral assay, and during this period, mice were allowed to habituate in the room where the assay would be carried out. Notably, we used doses of memantine similar to those from studies in other mouse models of neurodevelopmental disorders, which are expected to result in reversible blockade of a relatively small fraction of synaptic NMDA receptors[21,30].

**Ex vivo slice preparation**. All steps of electrophysiological experiments, including data collection and analysis, were performed by a researcher blinded to genotype. Acute hippocampal slices were prepared from mice 9 to 12 weeks of age. Animals were anesthetized with isoflurane and transcardially perfused with ice-cold oxygenated (95% $O_2$, 5% $CO_2$) cutting artificial cerebrospinal fluid (aCSF) solution (comprised of (mM): 2.5 KCl, 1.25 $NaH_2PO_4$, 5 $MgSO_4$, 0.5 $CaCl_2$, 200 sucrose, 25 $NaHCO_3$, 25 glucose, ~300 mOsm, 7.2–7.4 pH). After decapitation, brains were removed for sectioning in the same ice-cold cutting aCSF using a Vibratome (Leica Microsystems 1200 s). For whole-cell patch-clamp recordings of mE/IPSCs and the measurement of intrinsic membrane properties, 350-μm transverse hippocampal sections were prepared. For all voltage-sensitive dye experiments, 400-μm transverse hippocampal sections were prepared. Slices were recovered in the same cutting aCSF solution at 32 °C for 30 min and transferred to an oxygenated room-temperature solution composed of 50% cutting aCSF and 50% regular aCSF (comprised of (mM): 125 NaCl, 2.5 KCl, 1.25 $NaH_2PO_4$, 10 glucose, 26 $NaHCO_3$, 2 $CaCl_2$, 1 $MgCl_2$, ~300 mOsm, 7.2–7.4 pH) for 30 min. Subsequently, slices were transferred to 100% regular aCSF at room temperature for an additional 30 min before recording. All recordings were performed at 34 °C.

**Miniature spontaneous excitatory postsynaptic currents (mEPSCs)**. A pipette internal solution comprised of (mM): 140 $KCH_3OSO_3$, 5 KCl, 0.5 EGTA, 1 $MgCl_2$, 10 HEPES, 5 MgATP, 0.25 NaGTP, ~292 mOsm, $E_{Cl} = -78.8$ mV was used. Pipettes 4–6 MΩ in resistance were pulled from borosilicate glass capillaries (World Precision Instruments, 1B150F-4) on a Sutter Instruments P-1000 pipette puller. Voltage-clamp traces 5 min in duration were recorded at a holding potential of −70 mV in the presence of 1 μM tetrodotoxin (Tocris). All recordings were conducted with access resistance of <20 MΩ, leak current of <100 pA, and an applied series resistance compensation of 80%. Cells that did not maintain these parameters for the duration of the recording were eliminated. Analysis of mEPSCs was performed using pCLAMP10 (Axon Instruments, Molecular Devices) using a variable-amplitude template method, generated from a stable recording of at least 50 mEPSC events. Each trace was first low-pass filtered at 1 kHz, and negative-going mEPSCs were detected using a template match threshold of 4, without fitting.

**Miniature spontaneous inhibitory postsynaptic currents (mIPSCs)**. In order to record mIPSCs while maintaining a hyperpolarized membrane voltage, a high-chloride pipette internal solution (comprised of (mM): 100 $CsCH_3O_3S$, 50 CsCl, 10 HEPES, 0.2 BAPTA, 3 KCl, 1 $MgCl_2$, 0.25 GTP-Tris, 2.5 creatine phosphate disodium, 2 MgATP, ~296 mOsm, $E_{Cl} = -23.7$ mV) was used. Pipettes 4–6 MΩ in resistance were pulled from borosilicate glass capillaries (World Precision Instruments, 1B150F-4) on a Sutter Instruments P-1000 pipette puller. Voltage-clamp traces 3 min in duration were recorded at a holding potential of −70 mV in the presence of 1 μM tetrodotoxin (Tocris) and 2 mM kynurenic acid (Sigma Aldrich). All recordings were conducted with access resistance of <20 MΩ, leak current of <100 pA, and an applied series resistance compensation of 80%. Cells that did not maintain these parameters for the duration of the recording were eliminated. Analysis of mIPSCs was performed using pCLAMP10 (Axon Instruments, Molecular Devices) using a variable-amplitude template method, generated from a stable recording of at least 50 mIPSC events. Each trace was first low-pass filtered at 1 kHz, and negative-going mIPSCs were detected using a template match threshold of 4, without fitting.

**Voltage-sensitive dye imaging**. Following recovery, each slice was bulk-loaded with 100 μL of a solution containing the voltage-sensitive dye di-2-ANEPEQ[49,50]

(JPW1114, Molecular Probes) diluted at 0.05 mg/mL in aCSF. Each slice was stained for 14 min, washed with aCSF, and imaged in an interface chamber using an 80 x 80 CCD camera recording with a 1 kHz frame rate (NeuroCCD; RedShirt Imaging). Illumination was provided by a 530 nm Green LED (Thor Labs, M530L3-C2). A filter set (Chroma Technologies 11007v2 wide Green) was used to allow excitation at 510–560 nm and collection of emitted fluorescence at a wavelength > 592 nm.

A concentric bipolar tungsten microelectrode was used for stimulation of the perforant path. Two successive stimuli were applied with an inter-stimulus interval of 200 ms. Two stimulation intensities (200 μA and 400 μA) were used. Imaging trials lasted a total of 1000 ms each, with a 20 s inter-trial interval. Interleaved trials without stimulation allowed for background subtraction.

All data analysis was performed in IGOR (Wavemetrics) on the average of 12 trials. Data were displayed as the change in fluorescence divided by the resting fluorescence ($\Delta F/F$). Regions of interest (ROIs) were drawn for the molecular layer of the dentate gyrus, the dentate granule cell layer, and the hilus. Like other dyes of the ANEPP family, di-2-ANEPEQ decreases in fluorescence upon membrane depolarization. To be consistent with electrophysiological conventions, depolarizing $\Delta F/F$ signals were displayed as upward signals (warmer colors) and hyperpolarization $\Delta F/F$ signals were displayed as downward signals (colder colors). Snapshots of VSDI represent an $80 \times 80$ pixel window, corresponding to an ~$1.7 \times 1.7$ mm field of view.

**Postsynaptic density protein purification.** We adapted a widely used protocol for the preparation of PSD fractions[51]. Mice were sacrificed via cervical dislocation, and their brains quickly extracted. Forebrain cortical tissues were microdissected and Dounce-homogenized in 10 mL of homogenization buffer (0.32 M sucrose, 4 mM HEPES, pH 7.4 with protease inhibitors). The homogenate was centrifuged at $1000 \times g$ for 10 min at 4 °C to pellet cellular debris and nuclei (P1), and the subsequent supernatant (S1) was centrifuged for another 15 min at $10,000 \times g$ at 4 °C. The resulting pellet (P2, "crude" synaptosomes) was resuspended in another 10 mL of homogenization buffer and centrifuged at $10,000 \times g$ for 15 min at 4 °C. The supernatant was discarded, and the resulting pellet (P2') was resuspended in 10 mL of 4 mM HEPES (pH 7.4) then homogenized on ice. The lysate was incubated at 4 °C for 30 min while shaking to hypo-osmotically lyse the synaptosomes, and then centrifuged for 20 min at $25,000 \times g$ at 4 °C. The pellet (LP1) was resuspended in 1 mL of homogenization buffer and layered on top of a discontinuous sucrose gradient (bottom to top: 1.5 mL of 1.2 M sucrose, 1 mL of 1.0 M sucrose, and 1 mL of 0.8 M sucrose). The gradient was ultracentrifuged at $150,000 \times g$ for 1.5 h at 4 °C. The turbid layer between the 1.0/1.2 M sucrose interphase containing the synaptic plasma membranes (~1 mL) was collected and resuspended in 5 mL of 4 mM HEPES to dilute out the sucrose. This fraction was ultracentrifuged again at $200,000 \times g$ for 30 min at 4 °C. The resulting pellet was resuspended in 1 mL of 50 mM HEPES with 2 mM EDTA (pH 7.4), and the membrane proteins extracted by adding Triton X-100 at a final concentration of 0.5% and incubating at 4 °C while rotating for 15 min. The proteins were centrifuged at $32,000 \times g$ for 20 min at 4 °C, and the resulting pellet resuspended in 75 μL of 50 mM HEPES with 2 mM EDTA.

**Brain microdissection.** To assess CDKL5 protein expression in various brain regions, adult male mice were sacrificed by cervical dislocation. After decapitation, brains were removed and sectioned into 1 mm coronal slices using a mouse brain matrix. Tissue was dissected from the somatosensory cortex, striatum, hippocampus, and cerebellum and homogenized in lysis buffer containing 1% NP-40, pH 8.0.

**Western blot.** Protein concentration was measured using a Bradford assay. Purified synaptic density membrane proteins or protein lysates were prepared for gel electrophoresis by adding 4X LDS Sample Buffer (NuPAGE, NP0008) to a final concentration of 1x and β-mercaptoethanol to a final concentration of 5%. Samples were heat-denatured at 75 °C for 20 min, and 7.5 μg of protein was loaded into each well of a 4–12% Bis-Tris gradient gel (Invitrogen 10-well, 1.5 mm; NP0335). Protein gels were run for 2 h at 125 V at room temperature on a ThermoFisher XCell SureLock mini-cell electrophoresis box (EI001) using BioRad PowerPac HC High-Current Power Supply (1645052), then transferred onto a nitrocellulose membrane (0.45 μm pore-size; Biorad 162–0115) for 1 h and 10 min at 27 V at room temperature. The resulting membrane was blocked with a 1:1 solution of LI-COR Odyssey blocking buffer (927–40100) and 1x PBS for 1 h at room temperature.

Primary antibodies used for CDKL5 detection are anti-N-terminal CDKL5[4] (in house; diluted 1:500), anti-GAPDH (MA5–15738, Invitrogen; RRID: AB_10977387; diluted 1:1000), anti-EB2 (AB45767, Abcam, diluted 1:1000). Secondary antibodies (Licor) are goat anti-rabbit IRDye 800CW and donkey anti-rabbit IRDye 680RD at dilutions of 1:10,000. Standard protocols were used for the Odyssey Infrared Imaging System for visualization and quantification.

For postsynaptic density studies, the primary antibodies used were anti-N-terminal CDKL5, anti-GluN1 (ThermoFisher, OMA1–04010; diluted 1:1000), anti-GluN2A (Frontier Institute, AB_2571605; diluted 1:200), anti-GluN2B (Frontier Institute AB_2571761; diluted 1:200), anti-GluA1 (Abcam, ab31232; diluted

1:1000), anti-GluA2 (Abcam, ab133477; diluted 1:2000), and anti-ß-Actin (Abcam, ab8226; diluted 1:10,000). Secondary antibodies (LI-COR) were goat anti-rabbit IgG IRDye800CW and goat anti-mouse IgG IRDye680LT at dilutions of 1:10,000, and incubated for 1 h at room temperature. Standard protocols were used for the Odyssey Infrared Imaging System (LI-COR) for visualization and quantification.

**Statistical analyses.** For behavioral assays involving Dlx-cKO mice, we chose similar sample sizes for all behavioral experiments based on previous published studies of *Cdkl5* constitutive knockout mice[4] and *Cdkl5* glutamatergic conditional knockout mice[14]. For behavioral assays involving drug administration, we used pilot behavioral cohorts to estimate effect sizes required for detecting significant wild-type-mutant differences and used similar samples sizes for saline- and drug-administered cohorts. Importantly, the number of mice used were pre-determined prior to the start of each experiment.

For behavioral assays, statistical analyses were performed using Prism (GraphPad). All data sets were analyzed using the Shapiro-Wilk test for normality. For one-sample comparisons, data sets with normal distributions were analyzed for significance using the one-sample *t*-test, whereas data sets with non-normal distributions were analyzed using the Wilcoxon signed-rank test. For two-sample comparisons, data sets with normal distributions were analyzed for significance using the unpaired Student's *t*-test, whereas data sets with non-normal distributions were analyzed using the Mann–Whitney test. Two-way repeated measures ANOVA or the Kruskal–Wallis test was conducted for the appropriate data sets with Holm-Sidak's multiple-comparison test. All one-sample, two-sample, and multiple-comparison tests were two-tailed.

All other assays that involved sub-sampling of animals were analyzed using R (The R Project for Statistical Computing). Each data set was analyzed using a linear mixed effect model, where *Genotype* was modeled as a fixed effect term and *Animal* was modeled as a random effect term. This model accounts for both between-animal and between-cell variation. For each assay, null and alternative models were constructed using the *lmer* function in the lme4 package[52] in the following format:

$$m0 = lmer(\text{Outcome} \sim (1|\text{Animal}), \text{REML} = \text{TRUE})$$

$$m1 = lmer(\text{Outcome} \sim \text{Genotype} + (1|\text{Animal}), \text{REML} = \text{TRUE})$$

For data sets involving a third term (e.g., cumulative frequency bin), the following null and alternative models were constructed, in order to test the significance of an interaction between Genotype and the third term:

$$m0 = lmer(\text{Outcome} \sim \text{Bin} + \text{Genotype} + (1|\text{Animal}), \text{REML} = \text{TRUE})$$

$$m1 = lmer(\text{Outcome} \sim \text{Bin} * \text{Genotype} + (1|\text{Animal}), \text{REML} = \text{TRUE})$$

To make statistical comparisons, the *KRmodcomp* function from the pbkrtest package (Halekoh and Højsgaard, 2014) was used:

$$KRmodcomp(m0, m1)$$

The KRmodcomp function reports a modified *F*-test statistic based on the Kenward and Roger approximation, which accounts for the small sample sizes in our study, modified numerator and denominator degrees of freedom, and a *p*-value. The estimated effect of Genotype is obtained from the alternative model constructed using the *lmer* function from lme4.

Post-hoc testing for linear mixed effect models (e.g., at cumulative frequency bins) was performed using the least-squares means method for multiple comparisons. The *lsmeans* package was used[53] on the alternative model generated by *lmer*:

$$lsmeans(m1, \text{pairwise} \sim \text{Genotype}|\text{Bin}, \text{mode} = \text{'kenward} - \text{roger'})$$

For the analysis of cumulative distributions (mEPSC and mIPSC inter-event intervals and amplitudes), all samples from each individual cell was sorted, binned, and averaged at percent intervals, effectively generating a binned cumulative distribution curve for each cell. Data from all cells of a given genotype were plotted at these distinct binned percent intervals with the mean and error bars indicating s.e.m. The results were analyzed using linear mixed effect models, incorporating a third term, cumulative frequency bin.

All graphs are plotted using Prism (GraphPad). Boxplot limits indicate the minimum and maximum, and boxplot center line indicates the median. In our figures, *p*-values between 0.05 and 0.1 are shown explicitly, * is used to denote all $0.01 < p < 0.05$, ** for $0.001 < p < 0.01$, *** for $0.0001 < p < 0.001$, and **** for $p < 0.0001$.

**Reporting summary.** Further information on research design is available in the Nature Research Reporting Summary linked to this article.

## Data availability
The summary statistics for Figs. 1–7 and Supplemental Figs. 2–4 are provided as a Source Data file. Original full-scan western blots are included in Supplemental Fig. 6. Information for all antibodies used are provided in the Methods section. All other relevant data are available from the authors.

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

## Acknowledgements

This work was supported by funding from The Loulou Foundation (Z.Z.), International Foundation for CDKL5 Research (Z.Z.), the Intellectual and Developmental Disabilities

Research Center (IDDRC) at CHOP/PENN U54HD086984 (D.A.C. and Z.Z.), R01NS102731 (Z.Z. and D.A.C.), R01NS081054 (Z.Z.), R01NS038572 (D.A.C.), T32MH017168 (I.T.W.), F31NS101762 (B.T.), T32HD083185 (S.T.), T32GM007170 (S.T.), F30NS100433 (S.T.), and the Hearst Foundation Fellowship (S.T.). We thank the Preclinical Models Core of IDDRC at CHOP/PENN (U54HD086984, Timothy O'Brien, PhD) for providing training and resources.

## Author contributions

S.T., B.T., I.T.W., H.T., E.M., D.A.C., and Z.Z. designed the experiments and reviewed and interpreted the data. S.T., B.T., and I.T.W. performed the experiments with assistance from N.S. and K.S. Y.C. managed mouse colonies. S.T. wrote the manuscript with input from all other authors.

## Additional information

**Competing interests:** The authors declare no competing interests.

