## [Peer Review File · Nature Communications]

Reviewers' Comments:

Reviewer #1:

Remarks to the Author:

In this study, the authors convincingly show that loss of CDKL5 in GABAergic neurons results in ASD-like defects in mice (reduced social interactions, increased stereotyped behaviours). Interestingly, this phenotype is paralleled by E/I imbalance (increased glutamatergic transmission and postsynaptic NMDA receptor levels). The authors also generated a novel model of CDKL5 deficiency disorder (CDD) by knocking-in a patient-associated nonsense mutation of the kinase domain of CDKL5 (R59X mice). These mice show increased NMDA receptor levels accompanied by ASD-like behaviours, which were reduced by the administration of low doses of the NMDA antagonist memantine. The authors conclude that increased NMDAR signaling and circuit hyperexcitability underlie ASD-like features in mouse models of CDD.

This is an interesting study, that in my opinion deserves publication in Nature Communication. However, there are several issues that the authors should address before the manuscript can be considered acceptable for publication.

- 1) My major concern regards the fact that the authors do not describe the effect of memantine in Dlx-cKO mice. Did they try these experiments? If not, I think these experiments would be important to strengthen the whole structure of the manuscript, assuming that memantine is able to reduce ASD-like defects also in Dlx-cKO mice. Conversely, if the authors already performed these experiments but obtained negative results (memantine does not ameliorate ASD deficits), this data should be seriously taken into consideration for the overall interpretation of the study results. In any case the authors must address this issue.
- 2) Do Dlx-cKO and R59X mice show spontaneous or induced seizures? This issue should at least be addressed in the discussion.
- 3) Figure 4. The increase in GluN1 band intensity in Dlx-cKO mice (Fig. 4A) is not evident, please provide a more representative image. Bands for the housekeeping reference protein (Actin) should also be provided in this figure (not only in Suppl. data).
- 4) Some ASD-relevant behavioural features of R59X mice are referred to as "not shown", while other features are reported in Figure 6. The data not shown should be reported or at least described.
- 5) Statistics. The text should report the exact p values (and F values for ANOVA); alternatively (depending on editorial policy) a statistical data table should be provided for all experiments.
- 6) Discussion page 10. When referring to the selective increase of GluN2B in R59X mice, the author state that this mirrors previous findings in Cdkl5 constitutive KO. They should also explicitly make clear that this defect is evident in Dlx-cKO mice.

Reviewer #2:

Remarks to the Author:

This paper represents the logic extension of a previous publication in which the authors demonstrated that the conditional ablation of Cdkl5 from mouse forebrain glutamatergic neurons mainly affect learning and memory. However, both CDKL5 patients and Cdkl5 null mice manifest autistic phenotypes and impaired sociability whose origin remained unknown. By conditionally removing Cdkl5 from GABAergic neurons they were able to demonstrate that autistic features primarily originate from GABA neurons in CDKL5 deficient disorder (CDD). They then proceeded revealing that the observed phenotypes are associated with hyperexcitability and increased levels of NMDAR. The translational relevance of these results is highlighted by the demonstration that these phenotypes are present in a novel Cdkl5R59X, mouse line that bears a very early pathogenic premature stop codon, therefore

mimicking the null animals. Further, autistic features are ameliorated by a molecule reducing NMDAR signaling.

This is a novel paper that, although being related to a rare disorder, will be of great interest and importance for the CDD field and the community working on autism spectrum disorders.

Generally, the whole work is convincing. Experiments are very well performed, statistical analyses are correct and the experimental procedures are adequately detailed (for exceptions see below). Further, the text is very easy to follow and it contains only minor mistakes that we suggest to correct.

Thus, although my general comment is highly positive, I would suggest some revisions.

In particular:

1) In the introduction, as it is the case for most of the Cdkl5 papers, it is not mentioned that transgenic animals have a quite subtle phenotype often leading the community to debate on the translational value of these models. Although I do believe that mouse models of Cdkl5 are appropriate models for studying these disorders, I do find important that papers include some comments on that. Readers have to be aware that the human disorder is much more severe than the clinical condition of the animals that have been generated so far; this might be given by the lack of seizures.

2) Authors should note and possibly comment whether contrasting data in the field might derive from the use of different transgenic mice or diverse genetic backgrounds and mating procedures (see lane 71).

3) Supplemental Fig. 1. In the legend, authors should provide the definition of abbreviated brain regions (panel A). Further, the presented cerebellum western blots manifest a Cdkl5 reduction that is not consistent with conclusions.

4) Concerning western blots, I am suggesting several revisions. It is not clear from the described procedures and the representative gels whether the 5 proteins have been immunodecorated on the same filter. Further, MWs and the acrylamide concentration are missing. The internal standard (actin) has to be presented. More than two samples have to be shown (as in supplementary Fig.5). Please note that representative lanes (just one for genotype) often do not exhibit the described phenotype. Eventually, I am not convinced that any conclusion can be drawn for GluN1. Authors should exhibit better western blots or at least a panel of several samples. Images have been cut too close to the positive band.

5) Glutamatergic and GABA Cdkl5 conditional mice manifest different circuit excitability posing the question on the state of the circuit in non-conditional mouse models of Cdkl5. However, no data has been generated on this topic. I would suggest adding this piece of information at least on male R59X animals.

6) Authors cannot state that they are certainly observing the postsynaptic fraction without checking the quality of their procedure. I did notice that they were not controlling it. I find this quite relevant because I believe memantine often acts on extra synaptic receptors. Can the authors exclude this possibility? Maybe this topic will benefit from a comment.

7) The conclusion of lanes 224-225 derives from the attenuation of the NMDA signaling. I suggest unifying in a single chapter the last two parts of results.

8) Please correct the CDKL5 nomenclature (human/mouse gene: CDKL5/Cdkl5; human/mouse protein: CDKL5/Cdkl5) and minor errors such as lane 60: is; 435: weeks.

Reviewer #3:

Remarks to the Author:

Tang et al. et al carried out conditional KO of Cdkl5 in cortical inhibitory neurons and found that the mice show ASD-like behavioral abnormalities, increased paired pulse facilitation in PP-DG synapses, increased mEPSC frequency at CA1 pyramidal neurons, increased NR2B and NR1 amounts in cerebral cortex postsynaptic density fraction. They further demonstrated Cdkl5 R59X KI mice show increased NR2B and amelioration of the ASD-like behavioral abnormalities in the R59X mice by acute

administration of memantine.

These results surely improved our understanding on the pathophysiology of CDKL5 deficiency disorder (CDD), a neurodevelopmental disorder characterized by epilepsy, intellectual disability, and ASD-like features. In particular, the contrast between the current results and previous results concerning excitatory neuron-restricted conditional KO of Cdkl5 well define the cellular components-associated with each symptom (i.e. excitatory neurons for intellectual disability and inhibitory neurons for ASD-like ones).

However, their claim that altered NMDAR signaling underlies ASD-like features may include some logical gaps. The mechanistic insights that contributes to better understanding of the basis of ASD-like behaviors are not clearly described.

Major concerns:

1. They combined the results of Cdkl5-CKO and R59X KI mice to conclude the altered NMDA receptor signaling underlies ASD-like features in these mutants. Firstly, they did not provided the sufficient behavioral results to consider the R59X as the Cdkl5 KO equivalent, describing that "Fig. 6 and data not shown". There still remains a possibility that the R59X involves some gain-of-function effects by the presumptive truncated N-terminal fragment. The memantine experiments were carried out by using R59X/y mice. The effects of memantine was clear in the reduction of grooming and social interaction (Fig. 6B, C). Did the authors performed similar experiments in KO or inhibitory neurons-CKO? In case of Y-maze, %alteration of WT seems to be decreased by the drug. Is it reasonable to say "it did not positively affect the learning and memory deficit in R59X mice?"

2. They described the "hyper excitability" of the mice. However, they showed the altered presynaptic form plasticity and the mEPSC frequency. They did not see the epilepsy or abnormal EEG. The rationale should be given why they focus the PP-DD synaptic plasticity and why they think the altered PPF reflects the hyper excitability.

3. Increased mEPSC frequency and unaltered mEPSC amplitude suggest an increment of the presynaptic release probability or increased number of functional synapses. Which mechanism underlies for their result? Authors presented two possibilities in Discussion. However, this point can be addressed by conventional immunostaining or EM analysis at least in part. There are no morphological results in this study. Readers may be left out of the mechanism that links the absence of Cdkl5 in inhibitory neurons and the altered synaptic properties including the increment of GluN1 or GluN2B. That may include an essential information for the molecular basis of ASD-like behavior in CDD.

4. How can we generalize their findings? Are there any other situations (not limited to CDD) in which autistic behaviors can be explained by the dysfunction of the inhibitory neurons? Including relevant references would be helpful for readers with a general interest.

Minor points

1. In Fig. 1A, what is meant by Soc and Non-soc in empty cage experiments?
2. In Fig. 2E, H, and K, including the statistical analysis to show the difference between the groups would be helpful to evaluate the results. Why do the authors focus the deviation from the value 1.0?
3. In Fig. 5B the GluN1 results may not be representative for the graph in 5E.
4. Validation of the CKO should include the double immunostaining.
5. Reference 4 lacks Journal title.

Reply to Reviewers

We thank all three reviewers for their thorough and constructive reviews, which described our work as “well performed,” “adequately detailed,” and “of great interest and importance for the CDD field and the community working on autism spectrum disorders.” Following their suggestions, we have conducted additional experiments to the best of our ability within the time frame and refined our interpretations where appropriate. In summary, we believe that our revised manuscript, incorporating the new data in Figure 5 and along with the rationale presented here, sufficiently addresses the concerns of the reviewers. Below is the point-by-point reply:

Reviewer #1

1) My major concern regards the fact that the authors do not describe the effect of memantine in Dlx-cKO mice. Did they try these experiments? If not, I think these experiments would be important to strengthen the whole structure of the manuscript, assuming that memantine is able to reduce ASD-like defects also in Dlx-cKO mice. Conversely, if the authors already performed these experiments but obtained negative results (memantine does not ameliorates ASD deficits), this data should be seriously taken into consideration for the overall interpretation of the study results. In any case the authors must address this issue.

We thank the reviewer for this suggestion and agree that this is an important experiment to support our conclusions. We have now conducted assays on Dlx-cKO mice using the same dose of memantine as for R59X mice. As shown in Figure 5 of the revised manuscript, we found that 1) saline-treated Dlx-cKO mice, in comparison to WT, show significantly increased repetitive grooming activity as well as diminished social interaction, and 2) acute memantine administration significantly ameliorated both the repetitive grooming phenotype and the social interaction deficit in Dlx-cKO mice. Taken together with the findings in the R59X mice, these results show that acute memantine is able to ameliorate the autistic-like features in multiple mouse models of CDD.

2) Do Dlx-cKO and R59X mice show spontaneous or induced seizures? This issue should at least be addressed in the discussion.

To date, we have not observed abnormal epileptic behaviors in our behavior cohorts of Dlx-cKO and R59X mice. This is similar to what was seen in the *Cdk15* constitutive knockout mice generated in Wang et al. 2012. Our interpretation is that despite the circuit-level hyperexcitability found in these mice, there are neural network differences between human and mouse brains that prevent CDKL5-deficient mice from having overt seizures at an early age. To clarify this issue, we are currently undertaking efforts to conduct long-term EEG records of our CDD mouse models at various ages and under different environmental conditions. We have addressed this point in the discussion section of this manuscript.

3) Figure 4. The increase in GluN1 band intensity in Dlx-cKO mice (Fig. 4A) is not evident, please provide a more representative image. Bands for the housekeeping reference protein (Actin) should also been provided in this figure (not only in Suppl. data).

We apologize for the poor quality of the GluN1 image in the original submission. We have included a more representative image for the GluN1 bands for WT and Dlx-cKO in the revised Figure 4. Unfortunately, the GluN1 antibody is one that we have had trouble with on western blots of postsynaptic density membrane preparations (the bands are not as sharp as the ones for other proteins). We have also included housekeeping reference proteins (Actin) for Figures 4 and 6.

4) Some ASD-relevant behavioural features of R59X mice are referred to as "not shown", while other features are reported in Figure 6. The data not shown should be reported or at least described.

We thank the reviewer for this suggestion and apologize for this ambiguity in the manuscript. We have adjusted the statement in our manuscript to discuss only the “autistic-like features” of the R59X mouse line.

5) Statistics. The text should report the exact p values (and F values for ANOVA); alternatively (depending on editorial policy) a statistical data table should be provided for all experiments.

As per editorial policy, we have now included a separate source data file that contains exact p values, F statistics for ANOVA, as well as additional summary statistics for each figure.

6) Discussion page 10. When referring to the selective increase of GluN2B in R59X mice, the author state that this mirrors previous findings in Cdkl5 contitative KO. They should also explicetely make clear that this defect is evident in Dlx-cKO mice.

We thank the reviewer for this suggestion and have revised the manuscript to include this point.

Reviewer #2

1)In the introduction, as it is the case for most of the Cdkl5 papers, it is not mentioned that transgenic animals have a quite subtle phenotype often leading the community to debate on the translational value of these models. Although I do believe that mouse models of Cdkl5 are appropriate models for studying these disorders, I do find important that papers include some comments on that. Readers have to be aware that the human disorder is much more severe than the clinical condition of the animals that have been generated so far; this might be given by the lack of seizures.

We thank the reviewer for this astute suggestion, and we have included in our revised discussion the differences between CDD mouse models and human symptomatology, including the lack of seizures. We also now mention the subtlety of the phenotypes of mouse models of CDD in the introduction.

2)Authors should note and possibly comment whether contrasting data in the field might derive from the use of different transgenic mice or diverse genetic backgrounds and mating procedures (see lane 71).

This is another excellent point, and we have revised our manuscript to mention the potential influence of genetic background on the differences among various CDD models.

3)Supplemental Fig. 1. In the legend, authors should provide the definition of abbreviated brain regions (panel A). Further, the presented cerebellum western blots manifest a Cdkl5 reduction that is not consistent with conclusions.

We have included an updated western blot comparing CDKL5 protein levels in the cortex and striatum of WT and Dlx-cKO mice, with clearer definition of brain regions (Supplemental Fig. 1). We are no longer comparing CDKL5 protein levels in the cerebellum, as levels are low in this brain region, and we believe that the comparisons in the cortex and striatum are sufficient to show cell type-specificity of the Dlx5/6-Cre driver.

4)Concerning western blots, I am suggesting several revisions. It is not clear from the described procedures and the representative gels whether the 5 proteins have been immunodecorated on the same filter. Further, MWs and the acrylamide concentration are missing. The internal standard (actin) has to be presented. More than two samples have to be shown (as in supplementary Fig.5). Please note that representative lanes (just one for genotype) often do not exhibit the described phenotype. Eventually, I am not convinced that any conclusion can be drawn for GluN1. Authors should exhibit better western blots or at least a panel of several samples. Images have been cut too close to the positive band.

We thank the reviewer for these helpful suggestions and apologize for any ambiguity in our original submission. For the blots in the supplemental figures (including Supplemental Fig. 5 and Fig. 6), each

pair of two lanes reflect a distinct WT and Dlx-cKO/R59X pair. For example, in Supplemental Fig. 5, our results reflect 4 distinct pairs of WT and R59X littermates. We have also included an updated Supplemental Figure 6, showing for each western blot: 1) any cuts made in the blot, 2) specific proteins probed on each cut section, 3) molecular weights, and 4) housekeeping reference protein. The acrylamide concentration is now included in the Materials and Methods section for post-synaptic density preparations. We have also made an effort to include wider cropping of the representative bands. For GluN1, we would like to point out that the quantitative difference between WT and Dlx-cKO is <25%, and therefore is likely subtle to detect by eye. However, because this difference is not seen in the R59X mice, we have refrained from making general conclusions about GluN1 across mouse models of CDD.

5) Glutamatergic and GABA Cdk15 conditional mice manifest different circuit excitability posing the question on the state of the circuit in non-conditional mouse models of Cdk15. However, no data has been generated on this topic. I would suggest adding this piece of information at least on male R59X animals.

This is a great point, and we have now included a discussion of the state of the circuit in constitutive loss-of-function CDD mouse models. Given the current data in the literature, we believe that the nature of E/I imbalance in R59X animals is likely a complex topic that requires investigation at multiple levels, using cellular, circuit, and behavioral assays. In our opinion, it is beyond the scope of this current study to investigate these questions, but we do believe it will be an interesting direction to pursue.

6) Authors cannot state that they are certainly observing the postsynaptic fraction without checking the quality of their procedure. I did notice that they were not controlling it. I find this quite relevant because I believe memantine often acts on extra synaptic receptors. Can the authors exclude this possibility? Maybe this topic will benefit from a comment.

We thank the reviewer for this comment. We adapted a widely used method for preparation of the postsynaptic density membrane fraction (Bermejo et al. *J Vis Exp.* 2014). We attach the following western blot, which shows progressive enrichment of GluA1 and PSD-95 in the PSD membrane fraction compared to the synaptosomal and cytoplasmic fractions.

The reviewer also raises an excellent point regarding the action of memantine on extra-synaptic receptors. Although our PSD preps are enriched for synaptic proteins, we cannot rule out that extra-synaptic NMDAR levels are also altered in our mouse models. Therefore, we have added a discussion regarding the potential extra-synaptic changes that may occur in Dlx-cKO and R59X mice, as well as the actions of memantine at extra-synaptic receptors.

7) The conclusion of lanes 224-225 derives from the attenuation of the NMDA signaling. I suggest unifying in a single chapter the last two parts of results.

We thank the author for this suggestion and have now combined the last two parts of the results in to a single section.

8) Please correct the CDKL5 nomenclature (human/mouse gene: CDKL5/Cdkl5; human/mouse protein: CDKL5/Cdkl5) and minor errors such as lane 60: is; 435: weeks.

Our understanding is that both human and mouse proteins are written all-upercase (e.g. CDKL5). We have derived this from the guidelines from MGI (<http://www.informatics.jax.org/mgihome/nomen/gene.shtml>).

Reviewer #3

Major concerns:

1. They combined the results of Cdkl5-CKO and R59X KI mice to conclude the altered NMDA receptor signaling underlies ASD-like features in these mutants. Firstly, they did not provided the sufficient behavioral results to consider the R59X as the Cdkl5 KO equivalent, describing that “Fig. 6 and data not shown”. There still remains a possibility that the R59X involves some gain-of-function effects by the presumptive truncated N-terminal fragment. The memantine experiments were carried out by using R59X/y mice. The effects of memantine was clear in the reduction of grooming and social interaction (Fig. 6B, C). Did the authors performed similar experiments in KO or inhibitory neurons-CKO? In case of Y-maze, %alteration of WT seems to be decreased by the drug. Is it reasonable to say “it did not positively affect the learning and memory deficit in R59X mice?”

We thank the reviewer for these excellent suggestions. We have now edited the statement to focus on the “autistic-like features” of the R59X mice. In addition, a truncated form of the CDKL5 protein product has not been detected on our western blots, likely a result of nonsense-mediated RNA decay. Thus, the R59X mouse is most likely a loss-of-function line.

We have now carried out memantine rescue experiments in Dlx-cKO mice. In our revised Figure 5, we show that memantine significantly ameliorates both the repetitive grooming phenotype and the social interaction deficit in Dlx-cKO mice, similar to the effect on R59X mice.

It is interesting that memantine appears to decrease the performance of WT mice on the Y-maze. We believe that this may be due to an optimal E/I balance achieved by WT mice, and that memantine perturbs this balance to impair learning and memory (either too little or too much NMDAR signaling is likely detrimental to learning and memory). If excessive NMDAR signaling were responsible for impaired learning and memory in R59X mice, memantine should have at least partially ameliorated this deficit. However, this was not the case, and we believe that the statement that “memantine did not positively affect the deficit in R59X mice” is a reasonable interpretation.

2. They described the “hyper excitability” of the mice. However, they showed the altered presynaptic form plasticity and the mEPSC frequency. They did not see the epilepsy or abnormal EEG. The rationale should be given why they focus the PP-DD synaptic plasticity and why they think the altered PPF reflects the hyper excitability.

We thank the reviewer for this astute comment. In using the VSDI assay, our primary intention was to measure the circuit excitability by assessing the magnitude of the VSDI response after a single stimulus. Surprisingly, we found no significant differences between WT and Dlx-cKO mice. However, via a paired-pulse paradigm, we were also able to measure, to an extent, frequency-dependent circuit activation. This showed that the circuits in Dlx-cKO are hyperexcitable to repeated stimuli. As the reviewer points out, PPF in this manner does not necessarily reflect constitutive circuit hyperexcitability, and therefore we are careful to only draw this interpretation in conjunction with the patch-clamp data showing enhanced mEPSC frequency in the Dlx-cKO mice. Accordingly, we have revised the results section to reflect this.

3. Increased mEPSC frequency and unaltered mEPSC amplitude suggest an increment of the presynaptic release probability or increased number of functional synapses. Which mechanism underlies for their result? Authors presented two possibilities in Discussion. However, this point can be addressed by conventional immunostaining or EM analysis at least in part. There are no morphological results in this study. Readers may be left out of the mechanism that links the absence of Cdk15 in inhibitory neurons and the altered synaptic properties including the increment of GluN1 or GluN2B. That may include an essential information for the molecular basis of ASD-like behavior in CDD.

We thank the reviewer for this suggestion. We agree that additional evidence, such as immunostaining or EM analysis, could provide insight into the mechanisms underlying the synaptic changes in mouse models of CDD. However, we believe that without a detailed and thorough investigation, the results may not provide significant insight and could be easily misinterpreted. For example, an increase in GluN2B level may result from 1) an increase in overall synapse density and/or 2) an increase in the number of GluN2B receptors per synapse. In the latter case, any of the processes of GluN2B synthesis, recycling, or trafficking between synaptic and extra-synaptic compartments may be affected. To explore these possibilities, one would need to accurately measure synapse number, assess synapse composition, and distinguish between synaptic and extra-synaptic populations of receptors. These experiments are difficult to accomplish, especially in brain slices, and we believe, are best undertaken as a separate study.

4. How can we generalize their findings? Are there any other situations (not limited to CDD) in which autistic behaviors can be explained by the dysfunction of the inhibitory neurons? Including relevant references would be helpful for readers with a general interest.

We thank the reviewer for this excellent suggestion. There are indeed multiple examples of mouse models where inhibitory neuron dysfunction lead to autistic-like features. We have included these relevant references in our revised manuscript.

Minor points

1. In Fig. 1A, what is meant by Soc and Non-soc in empty cage experiments?

We apologize for the confusion. The cylinders in Fig. 1A are empty, and we have adjusted the labels accordingly. We have also corrected a coloring scheme error in our original submission for Fig. 1A.

2. In Fig. 2E, H, and K, including the statistical analysis to show the difference between the groups would be helpful to evaluate the results. Why do the authors focus the deviation from the value 1.0?

We thank the reviewer for this astute comment. We focused on the deviation from the value 1.0 because during our pilot experiments, we found that at a stimulation intensity of 200 μ A and an inter-stimulus interval of 200 ms (as in Fig. 2), WT slices tended to show neither paired-pulse facilitation nor depression. We therefore conducted the assay with these parameters to maximize the sensitivity in detecting aberrant paired-pulse facilitation or depression in Dlx-cKO and Nex-cKO mice (i.e. PPF or PPD not observed in WT mice under the same conditions).

Following the reviewer's suggestion, we have conducted between-group comparisons, marking statistically significant findings with asterisks (Fig. 2 and Supplemental Fig. 3). The results support the conclusion that Dlx-cKO and Nex-cKO have contrasting profiles of excitability. We have also retained our one-sample *t* test results, as we believe they provide additional insight into the nature of circuit dysfunction in these mouse lines.

3. In Fig. 5B the GluN1 results may not be representative for the graph in 5E.

We thank the reviewer for this suggestion, and we have now incorporated a more representative sample to reflect the quantified results.

4. Validation of the CKO should include the double immunostaining.

This is a good point, and ideally, we would like to confirm cell type-specific knockout using immunostaining. Unfortunately, none of the currently existing antibodies for CDKL5 have this level of cellular resolution when applied to brain slices. We have therefore employed an alternative strategy using western blotting of microdissected brain regions to confirm cell type-specific knockout (Supplemental Fig. 1). This strategy has been used by several other studies that used the Dlx5/6-Cre driver to achieve conditional knockout of forebrain GABAergic neurons (Ohtsuka et al. PNAS. 2008; Baydyuk et al. PNAS. 2011)

5. Reference 4 lacks Journal title.

We have adjusted this reference.

Reviewers' Comments:

Reviewer #1:

Remarks to the Author:

I congratulate the authors for their excellent paper and good revisions.

I consider the paper acceptable for publication in its present form.

Yuri Bozzi

Reviewer #2:

Remarks to the Author:

The manuscript is now suitable for publication.

Reviewer #3:

Remarks to the Author:

The authors appropriately addressed some of my concerns. Although I expected their comments concerning the disease relevance from more broad point of view, their comment was limited to the CDD LOF mouse models.

Reply to Reviewers

We thank all three reviewers for their constructive reviews throughout this submission process, and are pleased to hear that they find the “paper acceptable for publication in its presentable form” and “suitable for publication” (Reviewers #1 and #2). We believe that our revised manuscript has now sufficiently addresses the concerns of all reviewers upon inclusion of Review #3’s comment regarding LOF mouse models (see below).

Reviewer #1

**no additional concerns

Reviewer #2

**no additional concerns

Reviewer #3

“The authors appropriately addressed some of my concerns. Although I expected their comments concerning the disease relevance from more broad point of view, their comment was limited to the CDD LOF mouse models.”

We appreciate Reviewer #3’s important point regard the broader relevance of our findings to the CDD community. We have now added a comment addressing this disease relevance in the ‘Discussion section’